# Revising the global biogeography of annual and perennial plants

Tyler Poppenwimer[1,2], Itay Mayrose[1✉] & Niv DeMalach[2✉]

There are two main life cycles in plants—annual and perennial[1,2]. These life cycles are associated with different traits that determine ecosystem function[3,4]. Although life cycles are textbook examples of plant adaptation to different environments, we lack comprehensive knowledge regarding their global distributional patterns. Here we assembled an extensive database of plant life cycle assignments of 235,000 plant species coupled with millions of georeferenced datapoints to map the worldwide biogeography of these plant species. We found that annual plants are half as common as initially thought[5–8], accounting for only 6% of plant species. Our analyses indicate that annuals are favoured in hot and dry regions. However, a more accurate model shows that the prevalence of annual species is driven by temperature and precipitation in the driest quarter (rather than yearly means), explaining, for example, why some Mediterranean systems have more annuals than desert systems. Furthermore, this pattern remains consistent among different families, indicating convergent evolution. Finally, we demonstrate that increasing climate variability and anthropogenic disturbance increase annual favourability. Considering future climate change, we predict an increase in annual prevalence for 69% of the world's ecoregions by 2060. Overall, our analyses raise concerns for ecosystem services provided by perennial plants, as ongoing changes are leading to a higher proportion of annual plants globally.

At the coarsest scale, terrestrial plants can be categorized into two main types of life cycles, annual and perennial[1,2]. Although crude, this categorization represents the most fundamental characteristic of plant species and illustrates the inherent trade-offs between reproduction, survival and seedling success[1,9]. Annual plants reproduce once and complete their life cycle within one growing season, whereas perennial plants live for many years and, in most cases, reproduce multiple times. The evolutionary trade-offs reflected in these strategies manifest in numerous functional attributes, such as leaf[10] and root[11] traits, invasiveness[12,13], genome characteristics[14] and community stability[15] and, therefore, have many consequences for ecosystem functioning and services[3,4]. For example, by allocating more resources belowground, perennials reduce erosion, store organic carbon, and have higher nutrient- and water-use efficiencies[4,16–18].

The differences between annual and perennial plants are noticeably reflected in agricultural settings. Despite being a minor part of global biomass[19], annual species are the primary food source of humankind, probably because they allocate more resources to seed output, thereby enhancing agricultural productivity. During the Anthropocene epoch, the global cover of annuals substantially increased because natural systems, often dominated by perennials, were converted into annual cropland[20,21]. Annual plants cover around 70% of the croplands and provide about 80% of worldwide food consumption[22]. Moreover, the proportion of annuals increases in many systems because woody perennials have a higher extinction rate[23], while invasive plant species tend to be annuals[12].

The annual life cycle has repeatedly evolved in at least 120 different families, suggesting that it provides a fitness advantage under certain conditions[24]. According to life-history theory, the optimal life cycle is determined by the ratio of seedling (or seed) survival to adult survival[25,26]. The reproductive mode of perennials requires multiple growing seasons[1], in contrast to annuals, which require only one growing season. Thus, any external condition that decreases the ability of plants to survive between growing seasons necessarily reduces the reproductive fitness of perennial species[25,26]. However, because annual species could survive such conditions as seeds rather than adults, their reproductive fitness may not be impacted[1]. Any condition that skews the survivorship ratio in favour of seeds should therefore increase the favourability of annuals. Consequently, annuals should be favoured when adult mortality is high and seed persistence and seedling survival are relatively high.

Numerous studies have discussed plant life cycles as primary examples of adaptation to different climatic conditions and provided estimates for their prevalence in various regions[5–8]. However, the data provided in many of these studies, which penetrated many current ecological textbooks[7,8], are problematic in several aspects. First, the current estimate for the global proportion of annual species (13%) is based on a century-old sample of merely 400 species[2], representing 0.1% of accepted plant species[27]. Second, current biome-level estimates are based on a single location and are extrapolated to represent the entire biome. For example, the desert biome is assumed to contain 42% annual species[6,7], an estimate that is based on data from the Death Valley

[1]School of Plant Sciences and Food Security, Tel Aviv University, Tel Aviv, Israel. [2]Institute of Plant Sciences and Genetics in Agriculture, The Hebrew University of Jerusalem, Rehovot, Israel. ✉e-mail: itaymay@tauex.tau.ac.il; Niv.demalach@mail.huji.ac.il

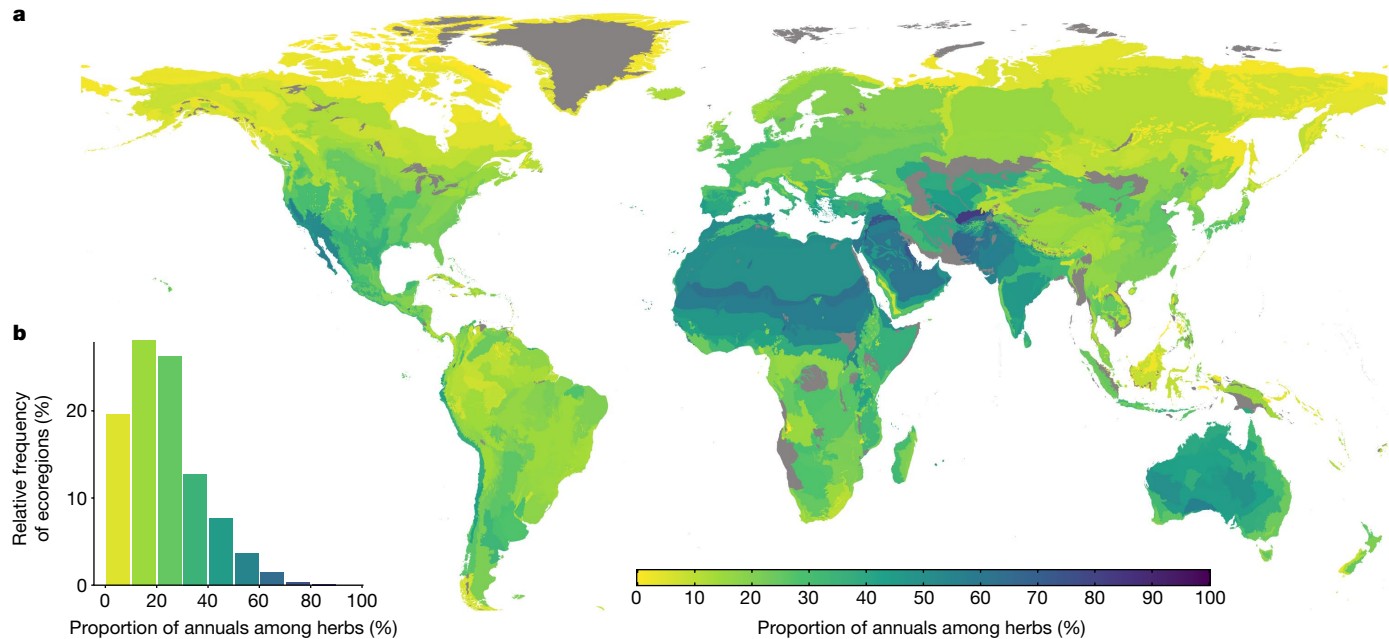

**Fig. 1 | The worldwide biogeography of annual plant prevalence. a**, A map of the proportion of annual species (among herbaceous species) in each ecoregion. **b**, The distribution of annual plant proportions among ecoregions. Ecoregions with insufficient data (Methods) are coloured grey, resulting in 682 coloured ecoregions.

in California only[2]. Third, estimates are inconsistent and difficult to compare due to ambiguous biome definitions. For example, an alternative estimate for the desert biome suggests that 73% of plant species are annuals[5,8]. Lastly, each biome incorporates a wide range of conditions, for example, the mean temperature in the desert biome ranges from 30° to −10 °C, corresponding to hot and cold deserts. Thus, this definition aggregates regions that differ markedly in their environmental conditions, probably affecting the prevalence of the different life cycles.

As central as life cycles are to plant ecology and evolutionary research, it is notable that we still have no precise estimate for the worldwide prevalence of life cycles and their environmental drivers. Yet, such an assessment is essential in times of climate and land-use changes[20,28], which are expected to substantially alter patterns of plant biogeography with many consequences for ecosystem processes and services[29–32]. Here we present a comprehensive assemblage of plant life-cycle data encompassing over 235,000 plant species. We cross this database with millions of georeferenced datapoints to produce the first worldwide map of plant life-cycle distribution. This extensive plant growth-form database contains life-cycle data for 67% of all vascular plant species and georeferenced data for 51%. These data enable us to evaluate the underlying drivers of plant life-cycle strategies of which testing is lacking at the global scale. We tested three key hypotheses, predicting that annuals are favoured under (1) increasing temperature and decreasing precipitation[24,33–35]; (2) high year-to-year variability in climatic conditions[35–37]; and (3) increasing human footprint (anthropogenic disturbance[36,38–40]). All of these hypotheses are based on the life-history theory that predicts annual species to be favoured with increasing adult mortality, relative to seedling mortality[25,26]. In other words, the relative abundance of annuals will be higher in regions with hot dry climates, high interannual variability and disturbance because they decrease adult survival. Finally, with a more accurate understanding of the global drivers, we provide an initial assessment regarding the impact of future conditions on plant life cycle distribution.

This compilation of life cycles revealed considerable differences in the relative prevalence of annual and perennial plant species compared with existing estimates. Annual species comprise 6% of all species and 15% of herbaceous species (that is, omitting all woody species, which are all perennial). Moreover, only 5.5% of ecoregions exhibit an annual herb proportion of 50% or more (Fig. 1).

Below, we focus on the proportion of annual species among herbaceous species (rather than among all species), which provides a better resolution in regions with a high proportion of woody species. Nonetheless, similar results were obtained when we analysed the proportion of annuals among all species (Supplementary Note 2, Supplementary Tables 7–12 and Extended Data Fig. 1).

The variation in the annual-herb frequencies across biomes supports the first hypothesis that annuals are favoured with increasing temperature and lower precipitation (Fig. 2). Still, the differences among biomes (according to a previously described approach[41]) were not substantial (Fig. 2a and Table 1). The proportion of annual herbs ranges from 13% to 25%, suggesting that the role of climate is underestimated at this coarse spatial scale. The large variability within each biome is revealed when examining the proportion of annuals at the ecoregion resolution (Fig. 2b). For example, not all desert-biome ecoregions have a high proportion of annual herb species, with cooler deserts exhibiting much lower proportions than hot ones. The same trend is repeated among other biomes, as ecoregions with lower precipitation and hotter temperatures (that is, located in the bottom left coordinate of their biome in Fig. 2b) possess a greater proportion of annuals.

This pattern was corroborated using a linear regression model that fitted the proportion of annuals as a function of mean yearly temperature and total yearly precipitation (Supplementary Table 1). These two climatic variables accounted for nearly half of the variance of the worldwide distribution of plant life cycle strategies ($P < 10^{-15}$, d.f. = 679, $R^2 = 0.48$). As mean yearly temperature increases and total precipitation decreases, the proportion of annuals increases (Fig. 2c). These results are robust to spatial autocorrelation, with only negligible differences in the parameter estimates and correspondingly low $P$ values (Supplementary Note 3 and Supplementary Tables 13 and 14) and to alternative statistical methods such as Poisson regression (Supplementary Note 4, Supplementary Tables 15–18 and Extended Data Fig. 2).

Although yearly temperature and precipitation provide a good description of annual herb proportions across the globe, it does not account for temporal variation in climate throughout the year. We

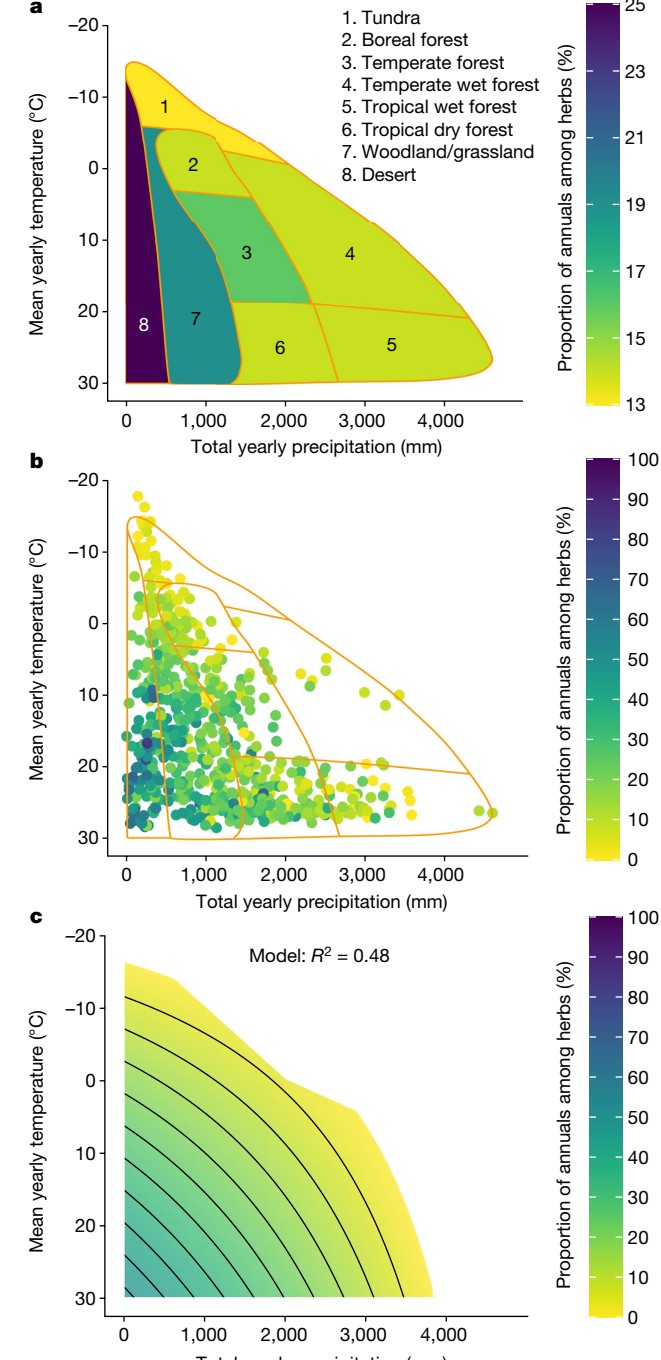

**Fig. 2 | The effects of total yearly precipitation and mean yearly temperature on the proportion of annuals (among herbaceous species). a**, The proportion of annuals in each of Whittaker's biomes[41]. **b**, The proportion of annuals in each ecoregion (the outline of Whittaker's biomes is marked by orange lines). **c**, Predictions of a linear regression model of the proportion of annuals as a function of the mean yearly precipitation and temperature (contour lines every 5%). Note that the scale is different for **a**. $n = 682$.

therefore fitted a suite of two-variable regression models. Each model consisted of one quarterly temperature variable and one quarterly precipitation variable. The best-fit model (hereafter, the quarterly model) incorporated the mean temperature of the warmest quarter and the log-transformed precipitation of the warmest quarter and accounted for 55% of the observed variance ($P < 10^{-15}$, d.f. = 679) (Supplementary Table 2). According to this model, annual herbs proportion increases with increasing temperature and decreasing precipitation

of the warmest quarter (Fig. 3). Furthermore, this model had a substantially better fit than the model based on the mean yearly temperature and total yearly precipitation outperforming it in terms of explained variance (0.55 versus 0.48) and information theory criteria ($\Delta$AICc = 92.4). These results provide a more nuanced understanding of the first hypothesis, demonstrating that hot and dry conditions impact the prevalence of annuals, particularly in the driest season.

The quarterly model can distinguish between ecoregions with similar yearly climate patterns yet a different proportion of annuals. For example, the eastern Mediterranean (for example, Tel Aviv) and Chihuahuan desert (southwestern USA and northern Mexico) ecoregions have identical mean yearly temperatures (17.6 °C) and relatively similar amounts of yearly precipitation (527 mm and 330 mm), yet maintain different annual herb proportions (51% and 36%). Given their similar mean yearly climate, the yearly temperature and precipitation model predicts similar annual herb proportions for these two ecoregions (32% and 34%, respectively). However, the Tel Aviv ecoregion receives substantially less precipitation (6 mm) than the Chihuahuan desert (157 mm) during the hottest quarter. As such, the quarterly model differentiates the two ecoregions, producing substantially better predictions (annual herb proportion of 52% in Tel Aviv and 29% in the Chihuahuan desert). Consequently, the coinciding of high-temperature and low-precipitation periods increases the favourability of annuals more than simply yearly means.

We conducted two analyses to account for potential biases of the revealed trends due to phylogenetic dependence. First, using the quarterly model, we conducted a separate analysis for the four most annual-rich families (Asteraceae, Brassicaceae, Fabaceae and Poaceae). Qualitatively similar relationships between climate and annual proportion were found in all families (Fig. 4), providing evidence for convergent evolution of annual life cycles in hot and dry conditions. We next tested the life cycle and climate relationship using phylogenetic generalized least squares (pGLS) analysis. We found that the median temperature of the warmest quarter for annuals is 3 °C higher, and the median precipitation of the warmest quarter is 35% lower (Supplementary Note 5 and Supplementary Tables 19 and 20). These results support the hypothesis that climate conditions during the driest period have a substantial role in driving the prevalence of annuals.

We tested the second hypothesis that increased year-to-year climatic variability favours annuals prevalence by focusing on interannual variability in total precipitation (in terms of the coefficient of variation) and mean temperature (in terms of s.d.). Using a bivariate regression, we found that increasing precipitation variability is associated with a higher proportion of annual species ($P < 10^{-15}$, d.f. = 679, $R^2 = 0.24$). Likewise, we found that increasing temperature variability also increases the favourability of annuals, although its effect is much weaker ($P = 0.0003$, d.f. = 679, $R^2 = 0.02$) (Supplementary Table 4). Furthermore, incorporating precipitation and temperature interannual variability into the quarterly model improved the model fit (from $R^2 = 0.55$ to $R^2 = 0.61$) (Supplementary Table 3) and overall performance ($\Delta$AICc = 51).

We examined our third hypothesis that increased human footprint (anthropogenic disturbance) should increase the proportion of annuals. In a bivariate regression, we found a positive effect of the human footprint on the proportion of annuals ($P < 10^{-7}$, d.f. = 680, $R^2 = 0.04$) (Supplementary Table 5). However, despite the mild individual explanatory power of human footprint, adding it to the previous model with the four climatic variables further improved the model's explanatory power ($\Delta$AICc = 15, change in $R^2$ from 0.61 to 0.63) (Supplementary Table 6).

Finally, we built a back-of-the-envelope projection of the expected prevalence of annuals in 2060 on the basis of predicted changes in the mean temperature and precipitation during the warmest quarter[42] (Extended Data Fig. 3). Under the simplifying assumptions that the prevalence of annuals in the future will follow the same climatic patterns without adaptation or time-lag, our model suggests that around 69% of ecoregions will experience an increase in the proportion of annuals.

**Table 1 | Previous estimates for the proportion of annuals among all species and among herbaceous species compared to our revised estimates**

| Region | Annuals among all species | | Annuals among herbaceous species | |
|---|---|---|---|---|
| | Previous estimate (%) | Revised estimate (%) | Previous estimate (%) | Revised estimate (%) |
| **Global** | 13 | **6** | 28 | **15** |
| Desert | 42 | **14** | 63 | **25** |
| Tundra | 2 | **11** | 3 | **13** |
| Woodland/grassland | 39 | **9** | 53 | **19** |
| Boreal forest | – | **11** | – | **14** |
| Tropical dry forest | – | **3** | – | **14** |
| Tropical wet forest | 16 | **3** | 44 | **14** |
| Temperate forest | 18 | **9** | 20 | **16** |
| Temperate wet forest | – | **7** | – | **14** |

Previous estimates are from ref. 7. Where indicated by a dash, there was no initial biome estimate. Alternative previous estimates are shown in Extended Data Table 2. Note that the biome nomenclature used for the previous estimates differs from ours and so the location of the original study was used to determine the corresponding biome. Further information is provided in Extended Data Table 3. Bold values indicate revised estimates.

## Conclusions

This study provides an extensive update to the worldwide biogeography of plant life cycles and demonstrates major differences compared with previous estimates. At the global level, our analyses indicate that annual species are half as common as previously thought[5–8]. Similarly, our estimates at the biome-level vary from earlier estimates, changing some by as much as three to fivefold (Table 1 and Extended Data Table 1). Moreover, these revised estimates display a more limited difference between the biome with the highest and lowest annual proportion, reducing the difference from 60% to a more restricted 12%.

Overall, our analyses provide general support for our three hypotheses regarding the conditions under which annual proportions will increase. First, we find that the proportion of annuals increases under hotter and drier conditions, and this result is robust to spatial autocorrelation and phylogenetic relatedness. However, yearly means provide an insufficient explanation for some observed patterns. After examining alternative climate patterns, we determined that a long dry summer is a principal factor governing the occurrence of annual-rich regions, demonstrating that the temporal distribution of hot and dry periods is more important than having an arid climate per se.

Second, our results suggest that annuals are more prevalent under increasing climate unpredictability. As interannual temperature variability increases and as interannual precipitation variability increases, the proportion of annuals also increases. However, the correlation between temperature variability and annual proportion is weaker than precipitation variability, indicating that irregular precipitation patterns have a greater impact.

Third, our findings demonstrate that, as human-mediated disturbance increases, the favourability of annual plants also increases. Furthermore, we found that a substantial portion of the effect of human disturbance is independent of climatic patterns. Although there is extensive evidence that human disturbance enhances the abundance of annuals in local communities[38], our study is one of the first to provide evidence that human disturbance favours annuals at the biogeographical scale.

Finally, our future projection model predicts that, by 2060, we will experience an increase in the prevalence of annuals. However, we caution that our back-of-the-envelope prediction is based on the simplistic assumption that biogeographical patterns instantly track climate changes (that is, it does not account for time lags in species response to a changing climate). Still, our prediction is also conservative in the sense that it does not account for the predicted increase in year-to-year climatic variability[42] as well as human footprint. With the human population predicted to reach 10 billion by 2060, anthropogenic activities are expected to have an increasing role in shaping patterns of plant biogeography. As a consequence, we expect a world with more annual-dominated ecoregions.

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

## Methods

### Life-cycle database development

We built an extensive life-cycle database by aggregating all types of vascular plant data from 11 disparate plant trait databases[14,43–52] (access dates are provided in Supplementary Note 6 and Supplementary Table 21). This raw database contained around 6.4 million entries and 400,000 unique names. All unique names were resolved using the R package WorldFloraOnline (v.1.7)[53] (WFO) to ensure a uniform naming scheme and to exclude unrecognized species. Resolved names were filtered by match distance and WFO acceptance (a full description is provided in Supplementary Note 7).

After name resolution, each entry consisted of a single species name and its associated trait term (for example, annual, forb/herb, tree, 10 years, shrub/herb, aquatic, tree, epiphyte). All unique trait terms were manually assessed to extract data relevant to a plant's life cycle (annual/perennial) and growth form (woody/herbaceous) when available. Those that did not provide relevant information (for example, Terrestrial_Trailing_Plant, 2.4, NO, b H) or provided conflicting information for the same entry were excluded (for example, shrub/herb, tree/terrestrial herb). After term interpretation, there were 5.6 million entries and 262,000 unique species remaining.

Life-cycle consensus among each species' data was achieved by comparing all life-cycle and biomass composition entries for that species. Only those species with a unanimous term agreement and without conflicting life-cycle and biomass composition consensuses were considered. Crop species were excluded as occurrence data may not represent natural habitats. A list of crop species was obtained from a previous study[14]. This process produced a database of 235,979 species with life cycle information. Our database contains 67% of all WFO-accepted plant species names and represents the largest plant life-cycle database assembled to date.

### Matching life-cycle data with species observations

Species observation data were based on occurrence data from the Geographic Biodiversity Information Facility[54] (GBIF). All observation datapoints within the Plantae kingdom (~355 million) were downloaded (14 September 2021) and processed locally. We filtered unreliable data points according to the recommendation provided in the vignette of the R package CoordinateCleaner (v.2.0-18)[55]. The following steps were used to filter unreliable data points:

(1) Datapoints without coordinates were excluded.
(2) The R package CoordinateCleaner (v.2.0-18)[55] was used to discard datapoints with the wrong locations and problematic temporal metadata (a full description of this process is provided in Supplementary Note 8).
(3) Datapoints were removed if the recorded 'coordinate uncertainty' was greater than 100 km.
(4) Datapoints of which the 'basis of records' was literature or living specimen were discarded (these generally refer to the location of museum or herbaria collections).
(5) Datapoints of which the record date was during or before 1945 were excluded as it has been suggested that these may be less reliable.
(6) Datapoints that were not labelled as species.

Once cleaned, all of the remaining unique names were resolved using the WFO package[53], and the same criteria as in the life form database were applied. Once the names were resolved, the species in the cleaned GBIF database and the assembled life form database were matched. Of the 235,979 species in our assembled lifeform database, 182,848 species were found within the cleaned GBIF data.

To mitigate sampling bias and inexact coordinates, species observation data were mapped into larger geographical regions defined by specific environmental and ecological conditions[14]. To this end, each georeferenced datapoint was assigned to one of 827 ecoregions as defined by the World Wildlife Fund[56] (WWF). This process was accomplished using the R packages raster (v.3.4-13)[57] and rgdal (v.1.5-27)[58]. According to previously described procedures[14], species were considered present in a geographical region only if there were five or more observations to ensure that the species had a sufficient established population. Similarly, to ensure that all regions contained sufficient data for analysis, each region was considered only if ten or more species were present.

This procedure produced sufficient data for 723 ecoregions when examining annual species among all species and 682 ecoregions for annual species among only herbaceous species.

We also analysed the data based on a grid system (using 100 km × 100 km cells) and found similar results to our main ecoregion-based analyses. Further details of these analyses are provided in Supplementary Note 9 and Extended Data Fig. 4.

### Predictors of annual proportion

We examined the relationships between various climatic and anthropogenic features and the distribution of plant life cycle strategies. To this end, we determined the frequency of plant life form strategies by considering the number of species with a given trait out of the total number of species with life-form data in each region (for example, annual species out of all species with life-cycle data). Each region was subsequently assigned a suite of climatic and anthropogenic features. Unless otherwise indicated, all features were determined by taking the median value across all pixels in a region.

We downloaded bioclimate features from the WorldClim Global Climate Data[59], which were developed from climate data during 1970–2000, at a ten arc-minute resolution. All 19 BIOCLIM variables representing each region's major temperature and precipitation characteristics were extracted using the R package raster (v.3.4-13)[57].

As a measure of climate unpredictability, we measured interannual precipitation variation (IPV) and interannual temperature variation (ITV). The IPV metric was obtained by extracting the coefficient of variation from ecoregion precipitation. The ITV metric was obtained by extracting the s.d. from the ecoregion temperature using the R package raster (v.3.4-13)[57]. We aggregated all available monthly precipitation/temperature data layers from the WorldClim Global Climate Data[59] at a ten arc-minute resolution (1961–2018) to determine the total yearly precipitation and mean temperature for each pixel in each year. The mean and coefficient of variation of the total yearly precipitation across all 58 years for each pixel were then used to determine IPV values. Similarly, the mean and s.d. of the yearly mean temperature across all years for each pixel were then used to determine ITV values (for temperature, the coefficient of variation is inappropriate because it inflates values near zero temperature). In contrast to other available bioclimatic features, such as BIOCLIM 3, which determine variability within a year, our estimates measure variability between years.

We used the Human Footprint data layer[60] as a proxy for anthropogenic disturbance, representing the human population's total ecological footprints. This layer incorporates eight variables: built-up environments, population density, electric power infrastructure, crop lands, pasture lands, roads, railways and navigable waterways. Together, these features evaluate the amount of land or sea necessary to support human activity's consumption habits. Human footprint values were extracted for each region using the R package raster (v.3.4-13)[57].

### Biome estimates

To obtain biome estimates of annual and annual herb frequencies, all ecoregions with sufficient data were individually plotted in the total yearly precipitation and mean yearly temperature space of the Whittaker biome overlay outline (adapted from ref. 41) (Fig. 2b). We determined each ecoregion's biome on the basis of its location within this space. For those ecoregions of which the biome designation was difficult to assess, their points were enlarged until one biome had a plurality of the circle's area. For those ecoregions outside Whittaker's

biome space, their biome designation was determined by the closest biome. Once the biome designation of all ecoregions was determined, the species presence data for all ecoregions within a given biome were aggregated. The same process used to determine the presence and absence of species in an ecoregion was used to determine the presence and absence of species in the biome. Notably, a biome could have more species than the combined ecoregions within said biome because some species may have five or more observations within the biome, but not within any of the individual ecoregions.

Whittaker's defined biomes were chosen to simplify comparisons to previous estimates (the terminology used in textbooks best matched those of Whittaker's definitions) and for their simplicity of using only temperature and precipitation.

## Comparing previous biome estimates

The classification approach for biomes used in previous estimates of annual proportions was not explicitly defined, making a direct comparison with our set of biomes difficult. However, we traced the origins of each estimate and determined the original study's locations. These locations were then matched with the WWF ecoregions, and the corresponding biome was determined as detailed above. This procedure enabled a direct comparison between previous estimates and our revised estimates. Moreover, previous studies did not explicitly provide estimates for the proportion of annuals among herbaceous species. Thus, for comparison purposes, previous annual herb proportion estimates were calculated on the basis of the biome-level life form classification estimates from each study. See Extended Data Tables 1–4 for the original study location matchings and annual herb proportion calculations.

## Statistical analyses

**Temperature and precipitation.** To assess support for our first hypothesis, we linearly regressed annual and annual herb frequency against mean yearly temperature, total yearly precipitation, and their interaction. Subsequently, we compared models based on two climatic variables, using one quarterly temperature bioclimatic variable and one quarterly precipitation variable, to identify a potentially better model. We used four temperature bioclimatic features (BIO8, BIO9, BIO10, BIO11) and four precipitation features (BIO16, BIO17, BIO18, BIO19). Month-specific bioclimatic features were omitted because they are highly correlated with quarter-specific features. Preliminary analysis suggested that log-transformations of precipitation bioclimatic features often increased explanatory power, and they were therefore also included in the exhaustive search.

Together, this grouping scheme produced 32 different two-feature linear regression models (four temperature and eight precipitation features) with an additional two linear regression models using mean yearly temperature and total yearly precipitation and the log-transformation of total yearly precipitation. Model comparison was achieved using AIC values obtained from the R package MuMIn (v.1.43.17)[61]. The best model was identified (hereafter the quarterly model) and then further applied to the four most annual-rich families (Asteraceae, Brassicaceae, Fabaceae and Poaceae).

## Climate uncertainty

To assess support for our second hypothesis, we investigated the role of IPV and ITV (proxies of climate uncertainty) on annual and annual herb frequencies. We began by testing each variable individually using linear regression and then tested whether their inclusion increased the fit of the quarterly model. Finally, we assessed the increased fit when both IPV and ITV were included in the quarterly model.

## Anthropogenic disturbance

To assess support for our third hypothesis, we measured the impact of the human footprint on annual and annual herb frequencies. We linearly regressed human footprint and annual/annual herb frequencies and then tested the change in model fit after its inclusion into the quarterly model with IPV and ITV.

## Phylogenetic biases

We used a pGLS analysis to account for phylogenetic dependence in the observed patterns. To this end, we devised a continuous response variable for each species by taking the median of the mean temperature of the warmest quarter and precipitation of the warmest quarter of all of their GBIF observations. The explanatory variable was a numeric conversion of each species' life cycle: 1 for annual and 0 for perennial.

The species were matched with those in the GBMB seed plant megaphylogeny constructed previously[62]. The same WFO name-resolution process was used on the species in the phylogeny to ensure the same naming scheme. Once matched, we selected the matching herbaceous species resulting in 20,819 species.

The results of the pGLS analysis were compared to the same model without the phylogenetic component (that is, standard linear regression) to assess the change in coefficient estimates and determine the overall impact of phylogenetic relatedness on our results (Supplementary Note 5).

## GBIF biases

To examine the biases in our dataset with regard to GBIF observational data, we linearly regressed annual and annual herb proportions against the $\log_{10}$-transformed total number of GBIF observations in an ecoregion. Similarly, we conducted a linear regression to assess the relationship between annual and annual herb proportion and the total number of present (5+ observations) GBIF species. Finally, we examined the species in GBIF, but missing from our dataset (Supplementary Note 10 and Extended Data Fig. 5).

## Spatial autocorrelation

According to previously described methods[63], spatial eigenvectors for our data were obtained using the R package adespatial (v.0.3.20)[64]. We selected the first set of eigenvectors (using those with both positive and negative eigenvalues) that accounted for at least 80% of the variance (39 eigenvectors) and incorporated them into the yearly-climate and quarterly-climate models. These results were compared to the same models without the eigenvectors included (Supplementary Note 3).

## Alternative regression models

To ensure that our results are robust to various regression methods, we applied two alternative regression methods. First, we applied a logit transformation to the proportion of annual herbs in each ecoregion followed by linear regression. Second, we used a generalized linear model (Poisson distribution with an offset to represent proportion data) (Supplementary Note 4 and Extended Data Fig. 2).

## Future projection

To obtain future projections of the proportion of annual herbs in each ecoregion, future climate estimates in the year 2060 were downloaded from the WorldClim Global Climate Data[59] using the 2041–2060, UKESM1-0-LL[65], ssp585, at a ten arc-minute resolution. The median values for each bioclimatic variable were extracted for each ecoregion using the R package raster (v.3.4-13)[57]. Using the coefficients of a linear regression between the two-most influential climatic parameters found in our study (mean temperature and precipitation during the warmer quarter, that is, the quarterly model) and their predicted median value in each ecoregion in 2060, we produced estimates for the proportion of annual herbs in each ecoregion with sufficient data. Year-to-year climate variability and human footprint were not incorporated due to data unavailability at the required resolution and scale. The projected annual herb proportion in each ecoregion was compared to its current estimate to determine the predicted change in proportion.

## Human footprint correlations

We examined the relationship between the human footprint and various bioclimatic features used in previous analyses. We assessed bioclimatic features 1 and 12 individually and together (the yearly model) and bioclimatic features 10 and 12 individually and together (the quarterly model) (Supplementary Note 12 and Extended Data Fig. 6).

## Reporting summary

Further information on research design is available in the Nature Portfolio Reporting Summary linked to this article.

## Data availability

All data are available at Figshare (https://doi.org/10.6084/m9.figshare.c.6176239).

## Code availability

All codes are available at Figshare (https://doi.org/10.6084/m9.figshare.c.6176239).

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

**Acknowledgements** We thank A. Rice for data-processing support and R. Milo for manuscript critique and feedback; M Mendel for statistical advice and support; D. Neves, R. Salguero-Gomez and B. Sandel for comments and critiques; and the staff at the Edmond J. Safra Center for Bioinformatics at Tel-Aviv University and the ISF for providing funding for this research. This work was supported by the Edmond J. Safra Center for Bioinformatics at Tel-Aviv University, and ISF grant no. 672/22 (to N.D.).

**Author contributions** Conceptualization: T.P., I.M. and N.D. Data curation: T.P. Formal analysis: T.P. Funding acquisition: T.P., I.M. and N.D. Investigation: T.P. Methodology: T.P. Project administration: I.M. and N.D. Resources: I.M. Software: T.P. Supervision: I.M. and N.D. Validation: T.P. Visualization: T.P. Writing—original draft preparation: T.P. Writing—review and editing: T.P., I.M. and N.D.

**Competing interests** The authors declare no competing interests.

**Additional information**
**Correspondence and requests for materials** should be addressed to Itay Mayrose or Niv DeMalach.

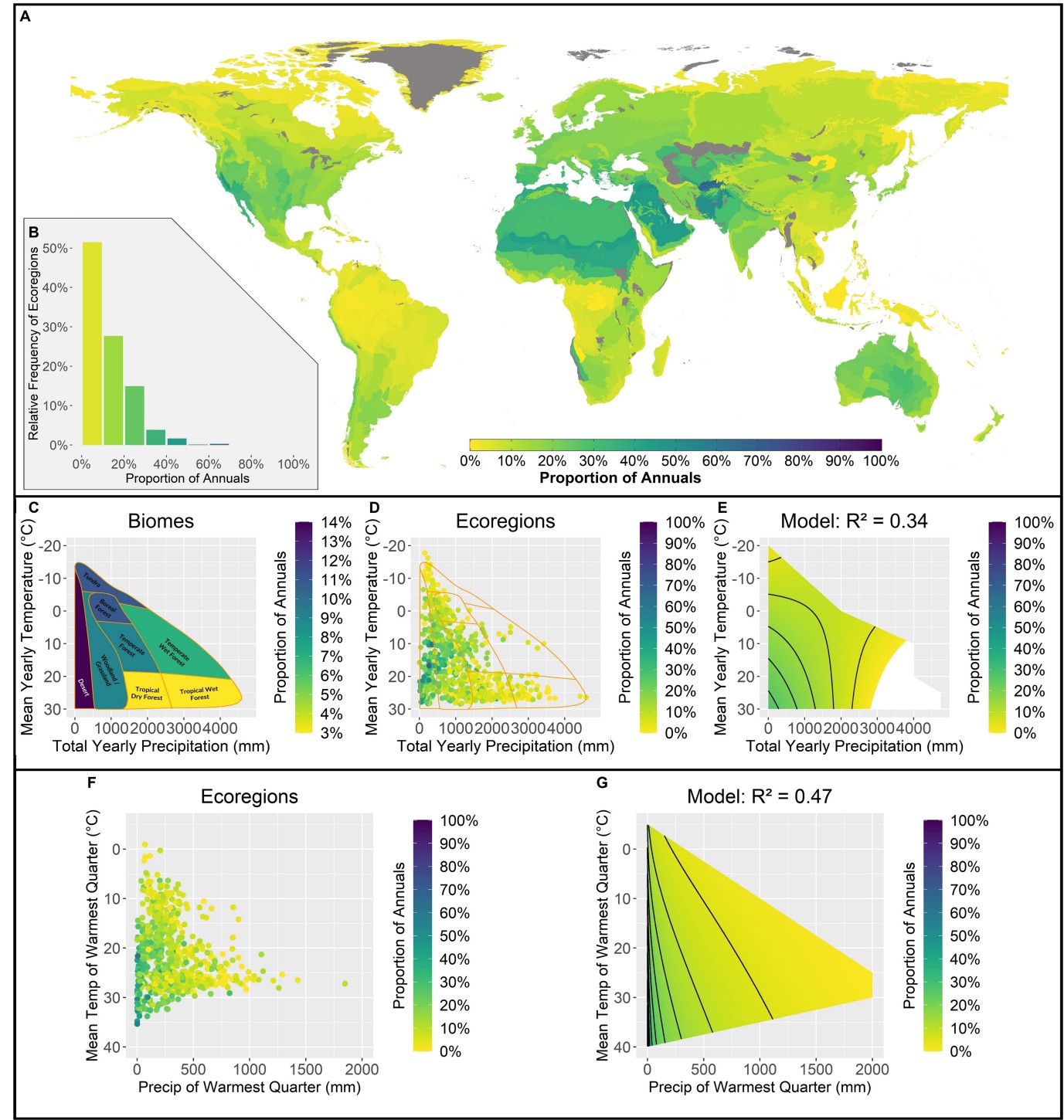

**Extended Data Fig. 1 | The global biogeography of the proportion of annual species and the effects of yearly and quarterly climate patterns. A**, the global proportion of annuals in ecoregions with sufficient data; ecoregions with insufficient data (see Methods) are coloured grey resulting in 723 coloured cells. **B**, The distribution of annual proportions among ecoregions. **C**, The annual proportions in each of Whittaker's Biomes. **D**, A scatterplot of the effects of mean yearly precipitation and temperature (the outline of Whittaker's biomes is marked by orange lines). **e**, Predictions of a regression model of the annual proportions as a function of mean yearly temperature and precipitation (contour lines every 5%). **F**, A scatterplot of the effects of total precipitation and mean temperature of the warmest quarter. **G**, Predictions of a regression model of the annual proportions as a function of total precipitation and mean temperature of the warmest quarter (contour lines every 5%). Note that the scale is different for panel **C**. N = 723.

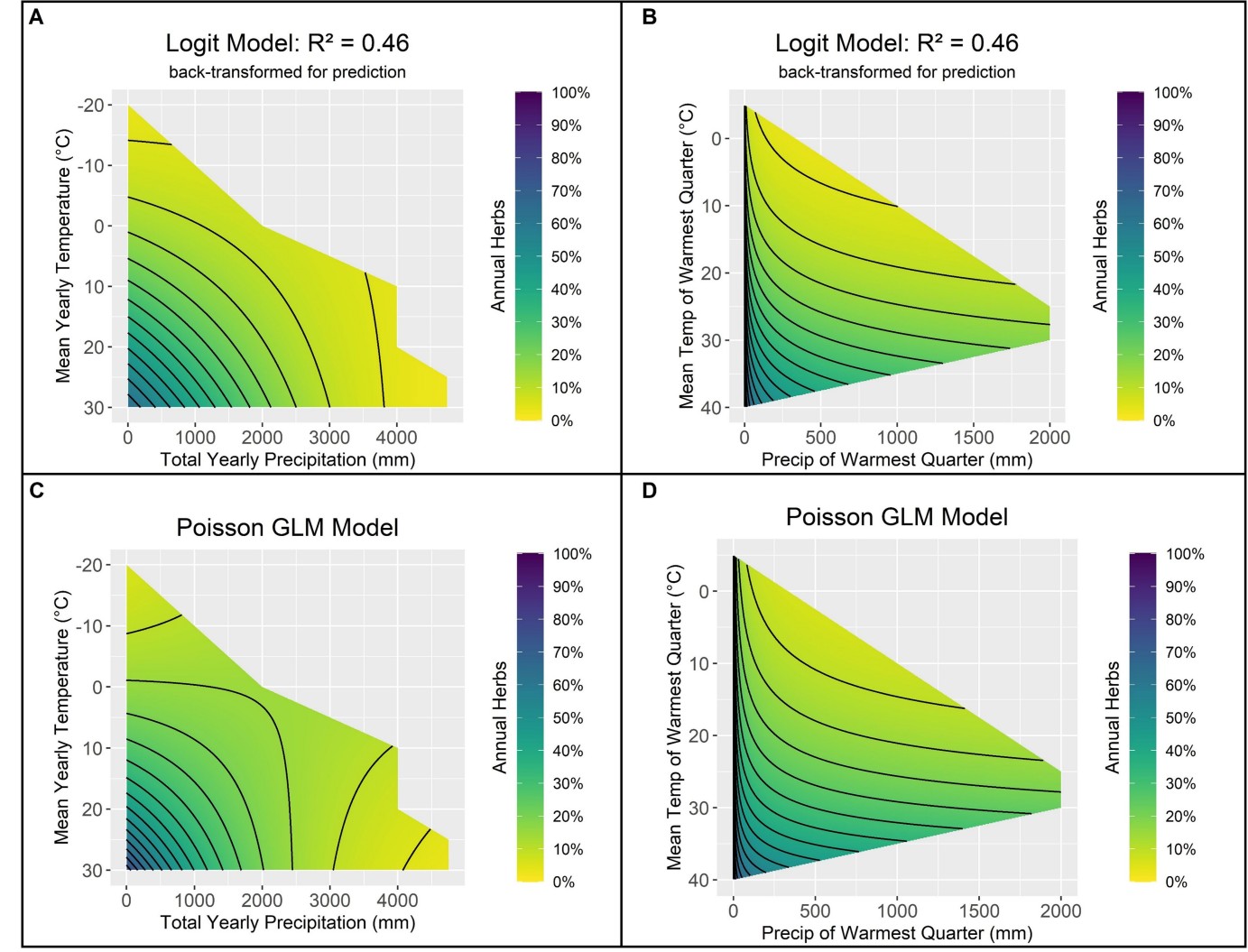

**Extended Data Fig. 2 | The predictions of alternative regression methods for the yearly and quarterly models. A** and **B** depict a logit transformation for the proportion of annual herbs, with **A** showing the yearly model and **B** showing the quarterly model. **C** and **D** depict the predictions of a Poisson Generalized Linear Model (GLM) with an offset ($\log_{10}$ of the species in an ecoregion) to obtain a rate for the proportion of annual herbs, with **c** showing the yearly model and **D** showing the quarterly model. Contour lines are every 5%.

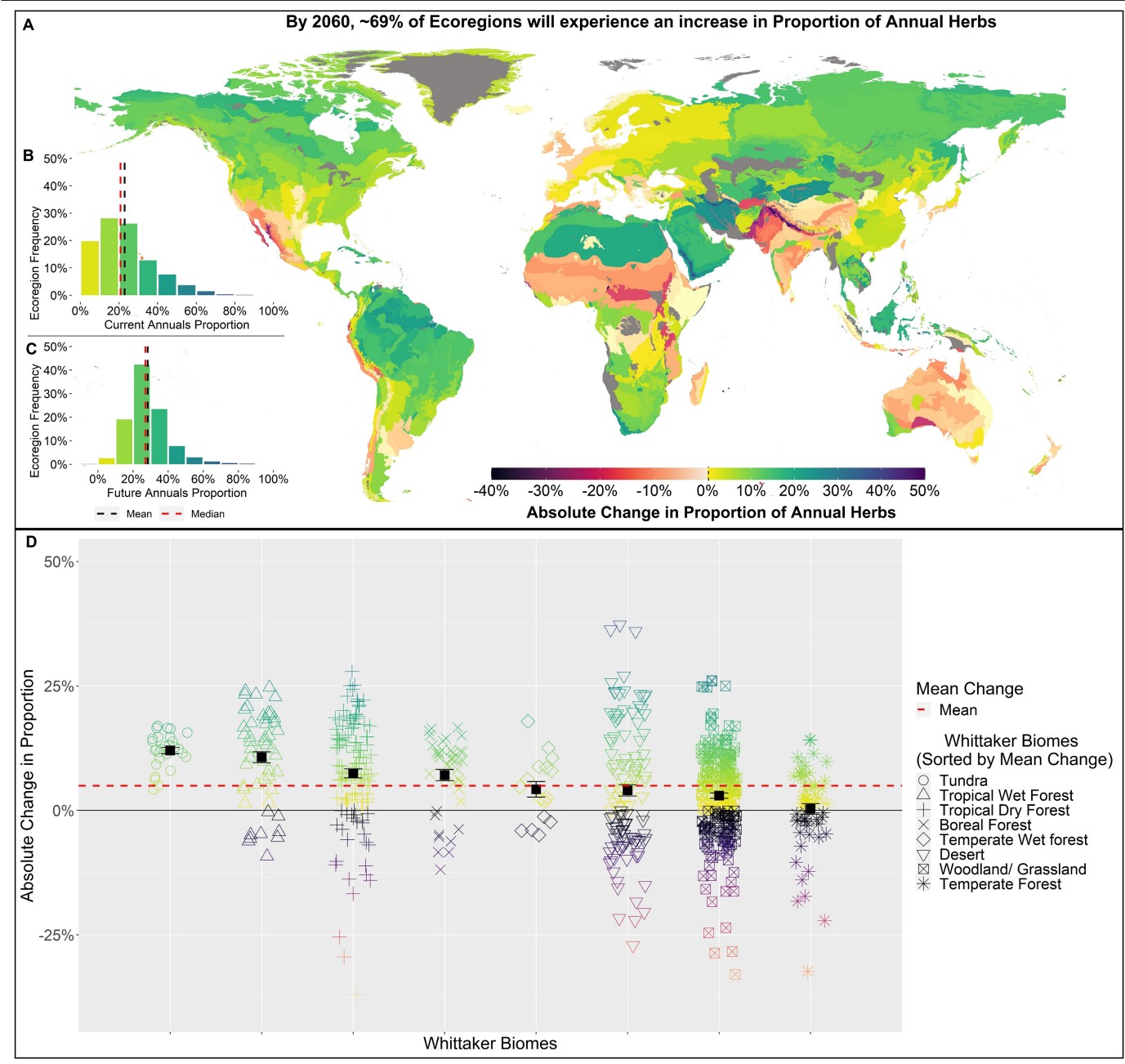

**Extended Data Fig. 3 | The predicted change in the proportion of annual herbs in 2100 based on expected climate patterns. A**, The absolute change in the proportion of annual herbs in ecoregions with sufficient data; those with insufficient data are coloured grey, resulting in 723 ecoregions. **B**, The current distribution of annual herbs proportions among ecoregions with the mean and median marked by vertical lines. **C**, The predicted future distribution of annual herbs proportions among ecoregions with mean and median demarcated. **D**, The absolute change in the proportion of annual herbs in ecoregions grouped by Whittaker Biome (the colour scale is the same as in **A-C**). N = 723.

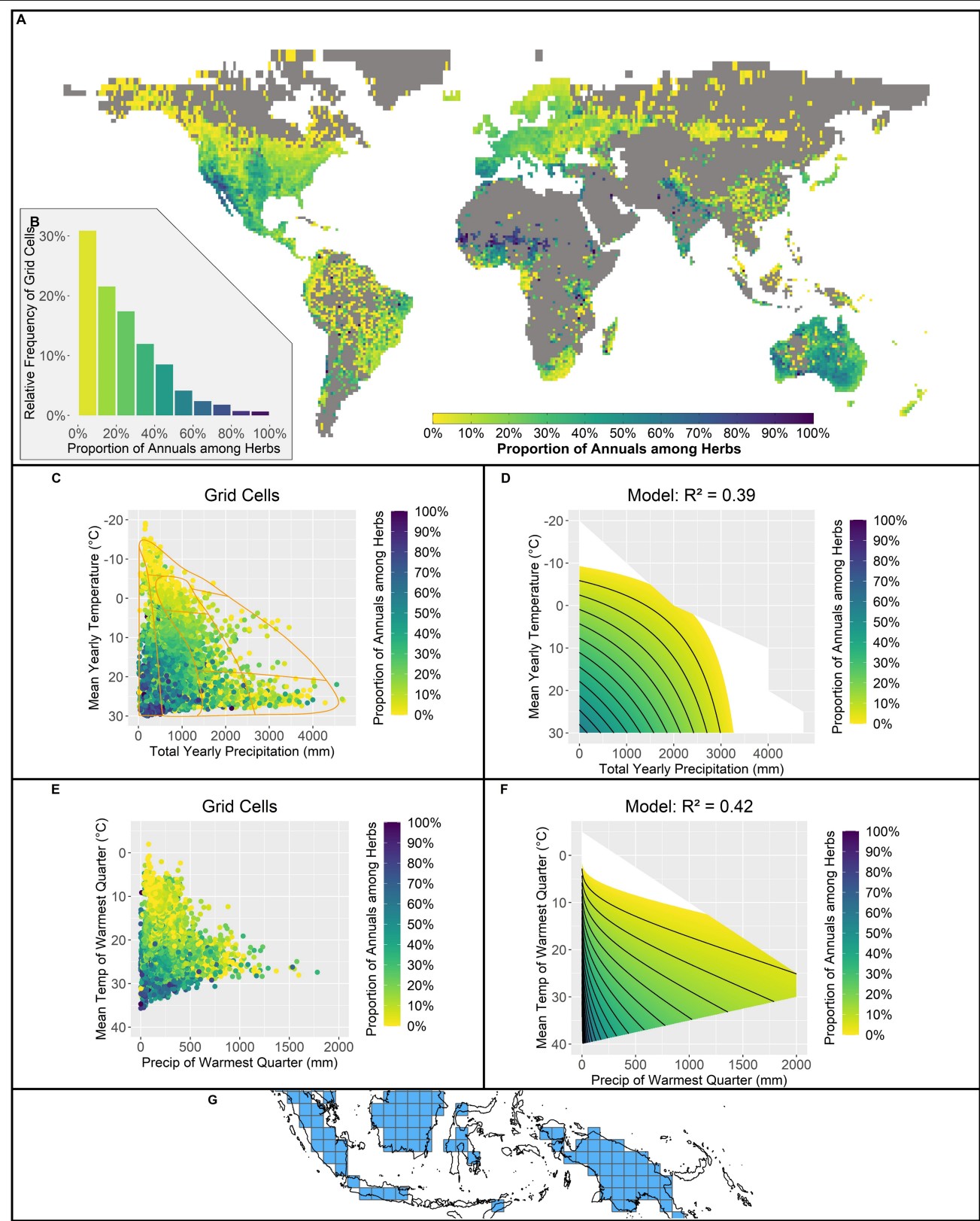

**Extended Data Fig. 4 | The global biogeography of the proportion of annual herbs and the effects of yearly and quarterly climate patterns using a gridded system. A**, The global proportion of annuals herbs in cells with sufficient data; cells with insufficient data (see Methods) are coloured grey resulting in 5,824 coloured cells. **B**, The distribution of annual herbs proportions among grid cells. **C**, A scatterplot of the effects of mean yearly precipitation and temperature (the outline of Whittaker's biomes is marked by orange lines). **D**, Predictions of a regression model of the annual herbs proportion as a function of mean yearly temperature and precipitation (contour lines every 5%). **E**, A scatterplot of the effects of total precipitation and mean temperature of the warmest quarter. **F**, Predictions of a regression model of the annual herbs proportions as a function of total precipitation and mean temperature of the warmest quarter (contour lines every 5%). N = 5,824. **G**, A sample of how enlarged grid cells (150 km × 150 km) omit islands/coastal regions.

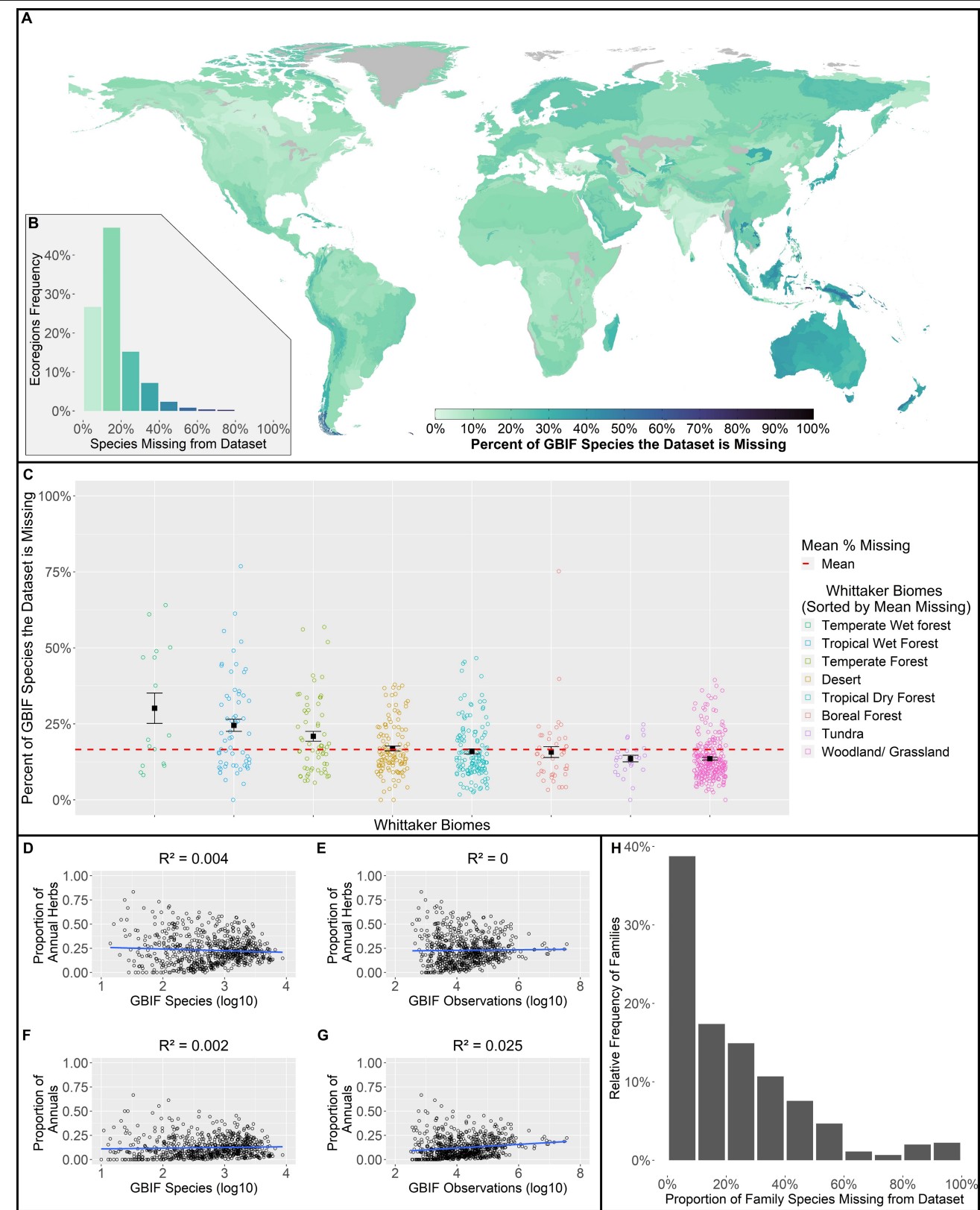

**Extended Data Fig. 5 | An exploration of the potential biases in the dataset.** **A-C** depicts the proportion of present GBIF species the dataset is missing, **A** as a global distribution map, **B** as a distribution of ecoregions, and **c** organized according to Whittaker Biomes. **D-F** shows the correlation of annual herbs (**D** and **E**) and annuals (**F** and **G**) with the number of GBIF species and observations. The blue lines depict the best-fit line. **H**, The distribution of the proportion of species in each family the dataset is missing.

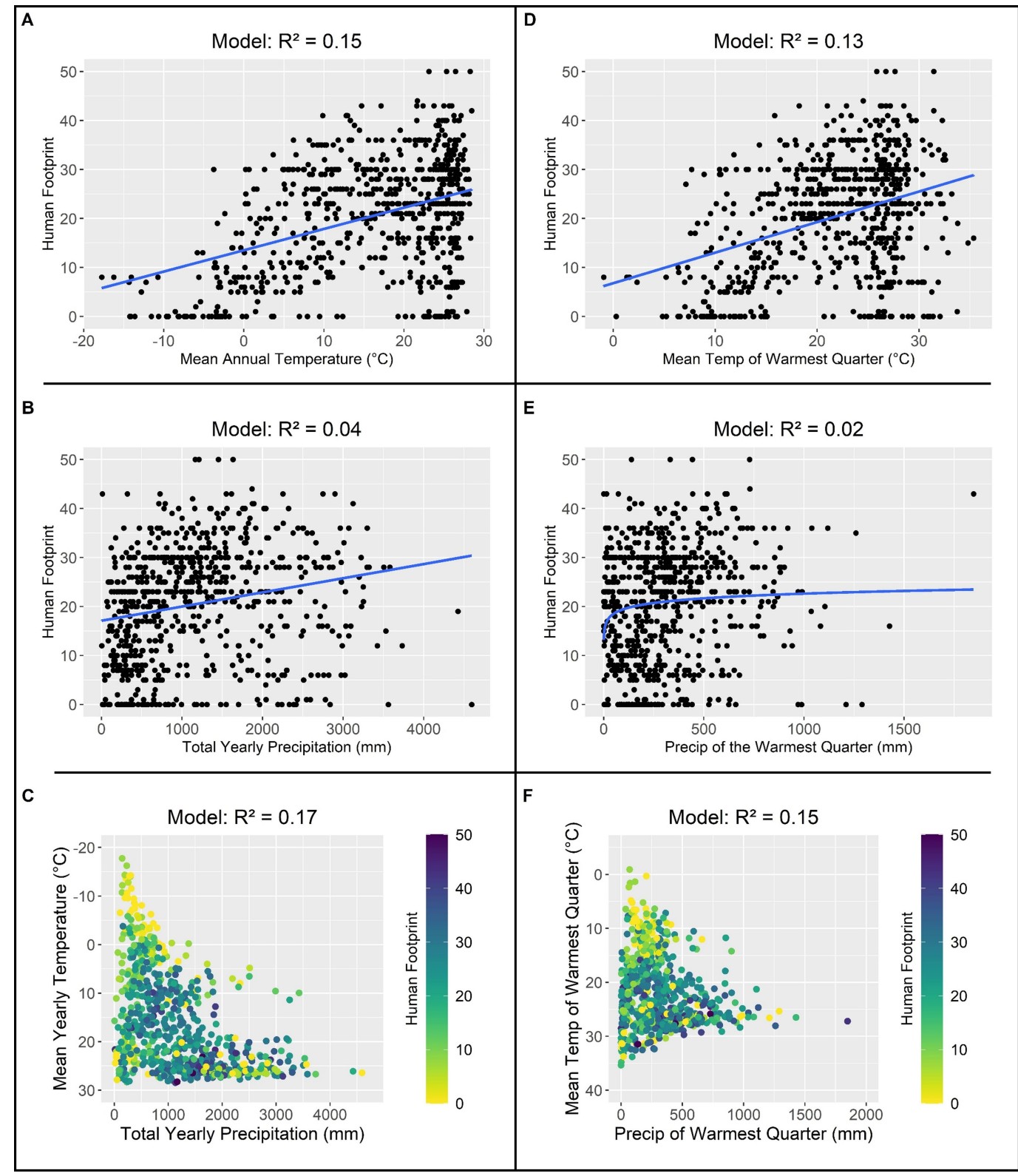

**Extended Data Fig. 6 | Scatter plots depicting the correlation between the Human Footprint and various BioClim Features. A-C** show the correlation of the Human Footprint with yearly temperature and precipitation individually **A, B** and together **C**. **D-F** show the correlation of the Human Footprint with quarterly temperature and precipitation individually **D, E** and together **F**. The blue lines depict the best-fit line. Note that the correlation in **E** and **F** are fitted to the log transformation of the quarterly precipitation.

**Extended Data Table 1 | A comparison of previous estimates, obtained from[4], for the proportion of annuals among all species and among herbs to our revised estimates**

| Region | Annuals among All Species | | Annuals among Herbaceous Species | |
|---|---|---|---|---|
| | Previous Estimate | **Revised Estimate** | Previous Estimate | **Revised Estimate** |
| **Global** | **13%** | **6%** | **29%** | **13%** |
| Desert | 73% or 27% | **14%** | 76%, 66% | **25%** |
| Tundra | 2% | **11%** | 3% | **13%** |
| Woodland / Grassland | 4%, 6%, 14% | **9%** | 11%, 13%, 16% | **19%** |
| Boreal forest |  | **11%** |  | **14%** |
| Tropical Dry Forest | 0% | **3%** | 0% | **14%** |
| Tropical Wet Forest | 10% | **3%** | 59% | **14%** |
| Temperate Forest | 7%, 2% | **9%** | 10%, 3% | **16%** |
| Temperate Wet Forest |  | **7%** |  | **14%** |

Greyed cells have no initial biome estimate. Alternative previous estimates from[3] are available in Table 1. Note, the biome nomenclature used for the previous estimates differs from ours and so the location of the original study was used to determine our corresponding biome. Additional information can be found in Extended Data Table 4.

**Extended Data Table 2 | The previous estimates obtained from[3,4], the location of the original study, our corresponding biome based on the original study location, and the calculations used to determine the proportion of annuals among herbs**

| Initial Biome Nomenclature | Original Study Location | Corresponding Biome Nomenclature | Annuals among All Species | Annuals among Herbaceous Species |
|---|---|---|---|---|
| **Begon & Townsend (2021)[3]** | | | | |
| Global | | Global | 13% | $\frac{13}{3+27+3+1+13} \approx 27.65\%$ |
| Arctic | Baffin's Island | Tundra | 2% | $\frac{2}{0+51+13+3+2} \approx 2.89\%$ |
| Desert | Death Valley | Desert | 42% | $\frac{42}{0+18+2+5+42} \approx 62.68\%$ |
| Tropical | Seychelles | Wet tropical forest | 16% | $\frac{16}{3+12+3+2+16} \approx 44.44\%$ |
| Temperate | Denmark | Temperate Forest | 18% | $\frac{18}{0+50+11+11+18} = 20\%$ |
| Mediterranean | The Camargue (mouth of the Rhone) | Woodland/ Grassland | 39% | $\frac{39}{0+23+11+39} \approx 53.42\%$ |

Annual herbs calculation: $\dfrac{Therophyte}{Epiphyte+Hemicryptophyte+Geophyte+Hydrophyte+Therophyte}$

| Initial Biome Nomenclature | Original Study Location | Corresponding Biome Nomenclature | Annuals among All Species | Annuals among Herbaceous Species |
|---|---|---|---|---|
| **Gurevitch _et al._ (2021)[4]** | | | | |
| Global | | Global | 13% | $\frac{13}{26+6+13} \approx 28.88\%$ |
| Tropical Rainforest | Queensland | Tropical Dry Forest | 0% | 0% |
| Subtropical forest | Matheran, India | Tropical Wet Forest | 10% | $\frac{10}{2+5+10} \approx 58.82\%$ |
| Warm-temperate Forest | Mediterranean live-oak forest, 0-500 m | Woodland /Grassland | 4% | $\frac{4}{24+9+4} \approx 10.81\%$ |
| Cold-temperate forest | Central Siskiyou Mtns | Temperate Forest | 7% | $\frac{7}{54+12+7} \approx 9.58\%$ |
| Tundra | Spitzbergen | Tundra | 2% | $\frac{2}{60+15+2} \approx 2.59\%$ |
| Mid-temperate mesophytic forest | Siskiyou Mtns South Fork and Beaver Creek | Temperate Forest | 2% | $\frac{2}{33+23+2} \approx 3.44\%$ |
| Oak Woodland | Santa Catalina Mountains, AZ | Woodland /Grassland | 6% | $\frac{6}{36+5+6} \approx 12.76\%$ |
| Dry Grassland | Pamir Mts. Steppe | Woodland /Grassland | 14% | $\frac{14}{63+10+14} \approx 16.09\%$ |
| Semi-desert | Oudjda semi-desert | Desert | 27% | $\frac{27}{14+0+27} \approx 65.85\%$ |
| Desert | Oudjda desert | Desert | 73% | $\frac{73}{17+6+73} \approx 76.04\%$ |

Annual herbs calculation: $\dfrac{Therophyte}{Hemicryptophyte+Cryptophyte+Therophyte}$

Calculations for annual herbs proportions from[3] were determined from values taken from the original studies. Calculations for annual herbs proportions from[4] were determined from values taken from the textbook (original study values were not available).

**Extended Data Table 3 | The original studies of global and biome-level annuals proportion estimates found in[6,7] their location and sample size**

| Biome | Proportion | Original Study | Location | Species |
|---|---|---|---|---|
| Global | 13% | Raunkiær, C. (1918). Über das biologische Normalspektrum. Andr. Fred. Høst & søn, Bianco Lunos bogtrykkeri. | Worldwide | 400 |
| Tropical | 16% | Raunkiær, C. (1918). Über das biologische Normalspektrum. Andr. Fred. Høst & søn, Bianco Lunos bogtrykkeri. | Seychelles | 258 |
| Desert | 42% | Raunkiær, C. (1918). Über das biologische Normalspektrum. Andr. Fred. Høst & søn, Bianco Lunos bogtrykkeri. | Death Valley | 294 |
| Mediterranean | 39% | Raunkiær, C. (1918). Über das biologische Normalspektrum. Andr. Fred. Høst & søn, Bianco Lunos bogtrykkeri. | The Camargue (mouth of the Rhone) | 233 |
| Temperate | 18% | Raunkiær, C. (1918). Über das biologische Normalspektrum. Andr. Fred. Høst & søn, Bianco Lunos bogtrykkeri. | Denmark | 1084 |
| Arctic | 2% | Raunkiær, C. (1918). Über das biologische Normalspektrum. Andr. Fred. Høst & søn, Bianco Lunos bogtrykkeri. | Baffin's Land | 129 |

Note that the origin for the global annual frequency estimate is the same as in Extended Data Table 4.

**Extended Data Table 4 | The original studies of global and biome-level annuals proportion estimates found in[5,8], their location and sample size**

| Biome | Proportion | Original Study | Location | Species |
|---|---|---|---|---|
| Global | 13% | Raunkiær, C. (1918). Über das biologische Normalspektrum. Andr. Fred. Høst & søn, Bianco Lunos bogtrykkeri. | Worldwide | 400 |
| Tropical Rainforest | 0% | Cromer, D. A. N., & Pryor, L. D. (1942). A contribution to rain-forest ecology. In Proceedings of the Linnean Society (Vol. 67, pp. 249-268). | Queensland, Australia | 141 |
| Subtropical Forest | 10% | Bharucha, F. R., & Ferreira, D. B. (1941). The biological spectrum of the Madras flora. Journal of University of Bombay, 9, 93-100. | Matheran, India | 361 |
| Warm-temperate Forest | 4% | Braun-Blanquet, J., (1936). La Chênaie d'Yeuse méditerranéenne (Quercion ilicis): monographie phytosociologique. Sta. Internatl. Dr Géobot Méditer. Et Alp. Montpellier Commun. **45**: 1-147 | Mediterranean live-oak forest, 0-500 m | |
| Cold-temperate Forest | 7% | Whittaker, R. H. (1960). Vegetation of the Siskiyou Mountains, Oregon and California. Ecological Monographs, 30(3), 279–338. | Central Siskiyou Mtns., by elevation belts on diorite: 1920-2140 m | 72 |
| Tundra | 2% | Raunkiær, C. (1918). Über das biologische Normalspektrum. Andr. Fred. Høst & søn, Bianco Lunos bogtrykkeri. | Spitzbergen | 110 |
| Mid-temperate Forest | 2% | Whittaker, R. H. (1960). Vegetation of the Siskiyou Mountains, Oregon and California. Ecological Monographs, 30(3), 279–338. | Mixed Evergreen Forest in the West (South Fork and Beaver Creek) | 160 |
| Oak Woodland | 6% | Whittaker, R. H., & Niering, W. A. (1965). Vegetation of the Santa Catalina Mountains, Arizona: a gradient analysis of the south slope. Ecology, 46(4), 429-452. | Santa Catalina Mountains, Arizona | 100 |
| Dry Grassland | 14% | Paulsen, O. (1915). Some remarks on the desert vegetation of America. The Plant World, 18(6), 155-161. | Pamir Mts. Steppe | 514 |
| Semi-desert | 27% | Braun-Blanquet, J., & Maire, R. C. J. E. (1924). Etudes sur la végétation et la flore marocaines. | Oudjda semi-desert | 32 |
| Desert | 73% | Braun-Blanquet, J., & Maire, R. C. J. E. (1924). Etudes sur la végétation et la flore marocaines. | Oudjda desert | 49 |

Note that the origin for the global annual frequency estimate is the same as in Extended Data Table 3.

# Reporting Summary

## Statistics

For all statistical analyses, confirm that the following items are present in the figure legend, table legend, main text, or Methods section.

| n/a | Confirmed | |
|---|---|---|
| ☐ | ☒ | The exact sample size (*n*) for each experimental group/condition, given as a discrete number and unit of measurement |
| ☒ | ☐ | A statement on whether measurements were taken from distinct samples or whether the same sample was measured repeatedly |
| ☐ | ☒ | The statistical test(s) used AND whether they are one- or two-sided *Only common tests should be described solely by name; describe more complex techniques in the Methods section.* |
| ☐ | ☒ | A description of all covariates tested |
| ☐ | ☒ | A description of any assumptions or corrections, such as tests of normality and adjustment for multiple comparisons |
| ☐ | ☒ | A full description of the statistical parameters including central tendency (e.g. means) or other basic estimates (e.g. regression coefficient) AND variation (e.g. standard deviation) or associated estimates of uncertainty (e.g. confidence intervals) |
| ☒ | ☐ | For null hypothesis testing, the test statistic (e.g. *F*, *t*, *r*) with confidence intervals, effect sizes, degrees of freedom and *P* value noted *Give P values as exact values whenever suitable.* |
| ☒ | ☐ | For Bayesian analysis, information on the choice of priors and Markov chain Monte Carlo settings |
| ☒ | ☐ | For hierarchical and complex designs, identification of the appropriate level for tests and full reporting of outcomes |
| ☐ | ☒ | Estimates of effect sizes (e.g. Cohen's *d*, Pearson's *r*), indicating how they were calculated |

*Our web collection on statistics for biologists contains articles on many of the points above.*

## Software and code

Policy information about availability of computer code

| Data collection | R version 4.1.1, R Studio version 2021.09.0 Build 351 R Packages: CoordinateCleaner v2.0-18, raster v3.4-13, rgdal v1.5-27 |
|---|---|
| Data analysis | R version 4.1.1, R Studio version 2021.09.0 Build 351 R Packages: raster v3.4-13, rgdal v1.5-27WorldFloraOnline v1.7, MuMIn v1.43.17 |

For manuscripts utilizing custom algorithms or software that are central to the research but not yet described in published literature, software must be made available to editors and reviewers. We strongly encourage code deposition in a community repository (e.g. GitHub). See the Nature Portfolio guidelines for submitting code & software for further information.

## Data

Policy information about availability of data

All manuscripts must include a data availability statement. This statement should provide the following information, where applicable:
- Accession codes, unique identifiers, or web links for publicly available datasets
- A description of any restrictions on data availability
- For clinical datasets or third party data, please ensure that the statement adheres to our policy

All data and code are available at: https://doi.org/10.6084/m9.figshare.c.6176239

# Research involving human participants, their data, or biological material

Policy information about studies with human participants or human data. See also policy information about sex, gender (identity/presentation), and sexual orientation and race, ethnicity and racism.

| | |
|---|---|
| Reporting on sex and gender | *Use the terms sex (biological attribute) and gender (shaped by social and cultural circumstances) carefully in order to avoid confusing both terms. Indicate if findings apply to only one sex or gender; describe whether sex and gender were considered in study design; whether sex and/or gender was determined based on self-reporting or assigned and methods used.*<br>*Provide in the source data disaggregated sex and gender data, where this information has been collected, and if consent has been obtained for sharing of individual-level data; provide overall numbers in this Reporting Summary. Please state if this information has not been collected.*<br>*Report sex- and gender-based analyses where performed, justify reasons for lack of sex- and gender-based analysis.* |
| Reporting on race, ethnicity, or other socially relevant groupings | *Please specify the socially constructed or socially relevant categorization variable(s) used in your manuscript and explain why they were used. Please note that such variables should not be used as proxies for other socially constructed/relevant variables (for example, race or ethnicity should not be used as a proxy for socioeconomic status).*<br>*Provide clear definitions of the relevant terms used, how they were provided (by the participants/respondents, the researchers, or third parties), and the method(s) used to classify people into the different categories (e.g. self-report, census or administrative data, social media data, etc.)*<br>*Please provide details about how you controlled for confounding variables in your analyses.* |
| Population characteristics | *Describe the covariate-relevant population characteristics of the human research participants (e.g. age, genotypic information, past and current diagnosis and treatment categories). If you filled out the behavioural & social sciences study design questions and have nothing to add here, write "See above."* |
| Recruitment | *Describe how participants were recruited. Outline any potential self-selection bias or other biases that may be present and how these are likely to impact results.* |
| Ethics oversight | *Identify the organization(s) that approved the study protocol.* |

Note that full information on the approval of the study protocol must also be provided in the manuscript.

# Field-specific reporting

Please select the one below that is the best fit for your research. If you are not sure, read the appropriate sections before making your selection.

☒ Life sciences  ☐ Behavioural & social sciences  ☐ Ecological, evolutionary & environmental sciences

For a reference copy of the document with all sections, see nature.com/documents/nr-reporting-summary-flat.pdf

# Life sciences study design

All studies must disclose on these points even when the disclosure is negative.

| | |
|---|---|
| Sample size | The sample size was the WWF-defined ecoregions with sufficient data (see Data Exclusions below) following the procedures used by Rice et al., 2019. It resulted in 723 ecoregions when examining the proportion of annuals among all plant species and 682 ecoregions when examining the proportion of annuals only among herbaceous species. |
| Data exclusions | Species-specific plant trait data were only included if the species name was recognized as accepted by World Flora Online and if all life cycle and biomass-composition trait information for a given species was in complete agreement.<br>Observation data were only included if 1) the observation had coordinates, 2) it was not flagged by the R Package CoordinateCleaner, 3) if the recorded 'coordinate uncertainty' was greater than 100km, 4) the 'Basis of Record' was literature or living specimen, 5) it's recorded data was after 1945, and 6) if it was labeled as species. This process followed the recommendation of the vignette of the R package CoordinateCleaner v2.0-18.<br>Ecoregion data was only included if 10 or more species were present and each of the 10 present species had 5 or more observations. This process followed the procedures used by Rice et al., 2019. |
| Replication | *Describe the measures taken to verify the reproducibility of the experimental findings. If all attempts at replication were successful, confirm this OR if there are any findings that were not replicated or cannot be reproduced, note this and describe why.* |
| Randomization | *Describe how samples/organisms/participants were allocated into experimental groups. If allocation was not random, describe how covariates were controlled OR if this is not relevant to your study, explain why.* |
| Blinding | *Describe whether the investigators were blinded to group allocation during data collection and/or analysis. If blinding was not possible, describe why OR explain why blinding was not relevant to your study.* |

# Reporting for specific materials, systems and methods

We require information from authors about some types of materials, experimental systems and methods used in many studies. Here, indicate whether each material, system or method listed is relevant to your study. If you are not sure if a list item applies to your research, read the appropriate section before selecting a response.

## Materials & experimental systems

| n/a | Involved in the study |
|-----|----------------------|
| ⊠ | Antibodies |
| ⊠ | Eukaryotic cell lines |
| ⊠ | Palaeontology and archaeology |
| ⊠ | Animals and other organisms |
| ⊠ | Clinical data |
| ⊠ | Dual use research of concern |
| ⊠ | Plants |

## Methods

| n/a | Involved in the study |
|-----|----------------------|
| ⊠ | ChIP-seq |
| ⊠ | Flow cytometry |
| ⊠ | MRI-based neuroimaging |

