## [Peer Review File · Nature]

Manuscript Title: Revising the global biogeography of plant life cycles

Reviewer Comments & Author Rebuttals

Reviewer Reports on the Initial Version:

Referees' comments:

Referee #1 (Remarks to the Author):

Poppenwimer et al. compile and explore a comprehensive dataset on the distribution of vascular plant species, combined with information on their life cycle strategies (annual vs perennial). The dataset is then used to address fundamental questions on the global biogeography of plant life cycle strategies, focusing on the importance of climate and disturbance factors in shaping variation on the proportion of annual plant species. While I find the manuscript interesting and with important results, there are major issues (listed below) that need to be dealt with before the manuscript is suitable for consideration in Nature.

Reasoning: The authors set out to test "key ecological hypotheses for plant life cycle strategies that were never tested on a global scale". Which hypotheses? Biomes represent a crucial component in the authors' analyses and results, but it is not clear how they fit into the narrative. What are the key ecological (and evolutionary) hypotheses for plant life cycle strategies across biomes? Should we expect similar patterns for similar biomes regardless of geography? Higher proportion of annual plant species reflect the evolutionary rarity of perenniality in biomes found under extreme climates?

Structure: Results and Discussion is almost entirely results, while the Conclusions section reads as Discussion. I'd suggest bringing/adapting most of the text in the Conclusions to the Results and Discussion section, and then have the Conclusions focused on the major questions of the manuscript. Also, as I pointed out above, the authors need to make clear in the Discussion which are the hypotheses being tested in the manuscript, and whether the results support or reject each of them.

Regressions: Have the authors checked for spatial autocorrelation (SAC) in model residuals? These are not trivial results, and should be reported accordingly. If the authors haven't checked for SAC, I'd suggest using (and presenting) correlograms, followed by (if SAC is present) generalised least squares (GLS) analyses with different spatial structures (e.g., spherical, exponential). Finally, the best model could be selected by comparing the delta AIC of each model.

Phylogenetic context: The authors try to address a potential artifact of phylogenetic dependence in the results by replicating some of the analyses using the four most annual-rich families. Why not formally test for phylogenetic dependence in, for example, a pGLS framework? The author could take advantage of published, comprehensive phylogenies (e.g. Zanne et al. *Nature*, 2014; Smith and Brown *AJB*, 2018) and then add missing taxa using imputation methods broadly used in the literature (e.g. SUNPLIN, randomized within the most recent common ancestor).

Biomes: Why use Whittaker's definition of biomes? Comparisons with previous studies is a minor component of the manuscript, and the authors seem concerned with ambiguous delimitations of terrestrial biomes (as expressed in the Introduction). Despite being climatically, compositionally and functionally distinct, many major biomes, such as savannas (the second largest tropical biome), are lumped within one of Whittaker's biomes. I'd suggest using Olson et al. (<https://academic.oup.com/bioscience/article/51/11/933/227116>) or Higgins et al. for a functional perspective (<https://doi.org/10.1111/gcb.13367>).

Spatial resolution: A key hypothesis in the manuscript is to test the importance of climatic and landscape factors in shaping the biogeography of plant life cycle strategies globally. Nonetheless, these factors are lumped into (very) coarse ecoregions, which encompass heterogeneous gradients of human disturbance and span multiple climatic domains. In the Cerrado ecoregion (South American savannas), for instance, climate spans from semi-arid in its northeastern portion to sub-humid, frost affected regions in the south. Why not assemble plant records into grid cells, which could then be classified into biomes according to their overlap with Olson's (or Higgins) delimitations. Issues with (i) grid cells being assigned to wrong biomes (delimitations are not perfect in heterogeneous landscapes, especially in dry regions of South America); (ii) low accuracy of geographic coordinates of plant records, or (iii) species misidentifications could be dealt with by applying randomizations and multiple iterations of the most relevant analyses.

Hope these comments are helpful.

Referee #2 (Remarks to the Author):

Overall thoughts:

The fundamental quantification of the frequency of annual lifecycles in plants, and its spatial variation is a useful product for a variety of purposes, ranging from evolutionary studies to vegetation modeling. In fact, it is so fundamental that I assumed that this had been done before. Some searching suggests that it has not, so this paper should fill an important gap.

My main critique is that the paper is fairly uncritical of the data used. There are numerous possible biases that could arise in the life cycle data and (especially) in the spatial data. These could have large effects on the estimated frequencies. My guess is that the biases are unlikely to overturn the major results here, but I think that needs some empirical support in the manuscript (some suggestions are below). And I think it very plausible that such biases could lead to errors in annual proportions of a few percentage points.

The statistical analysis of climate and annual proportion seemed mostly adequate, though I did not find it massively illuminating. I think the main novelty in the paper rests firmly on the map and biome-level patterns.

Concerns about biases:

The central goal of the paper – quantification of annual proportions – depends on species occurrences from GBIF. These data are not collected with the intent to obtain a representative sample of the life cycles within a region, or in fact with any standardized approach at all. Thus, there is a potential for important biases to arise, and these should be investigated. For example, individual plants may be sampled in GBIF approximately in proportion to their abundance rather than in proportion to the richness of their life cycle category. This will lead the more abundant group to be biased towards higher richness estimates. There might also be important biases towards sampling more conspicuous species (likely favoring perennials) and more consistently identifiable species (probably also favoring perennials). Such potential biases should be investigated.

Some useful diagnostics would include species accumulation curves for annuals and perennials within regions, and a plot of the relationship between number of GBIF records and proportion annual (e.g. as a region becomes better sampled, does it tend to “drift” towards higher or lower annual estimates)? Another approach that would help resolve incomplete (and crucially, differently incomplete between annual and perennial) sampling would be to apply a diversity estimator within each group within each region (such as the Chao2 estimator). These estimated richnesses would not be free from all kinds of biases, but at least some biases would be removed.

The compilation of life cycle data is impressive, but it would be helpful to give some more detail on its own potential biases. For example, what does the spatial pattern of missing data look like (that is, where are there regions with many species with GBIF records but no life cycle data?). How are the missing species distributed across the tree of life – are certain clades over- or under-represented and do these clades tend to be annual- or perennial-dominated?

Smaller comments:

Line 27 – I think the phrase “broadest scale” is not quite right here. This implies something about the scope or extent, whereas the main feature of this classification is its low resolution, or coarseness.

Line 38-44 – I think this section is interesting in general, but did not really understand its purpose here (especially when I later learned that crops were removed).

Line 59 – I have not seen this estimate (13% annual) before. I agree that it is old and seemingly based on too little data, but I’m not convinced that this is a widely accepted number without better more recent alternatives. Has there really been no other estimate in that time? For this section in general, I found the claims about the poor knowledge of annual proportions to be shocking – thinking that there must be better and more complete evidence already published. I did try to find such numbers and my search turned up little, so perhaps the situation is this bad, which certainly argues for the importance of this manuscript!

Line 187-189 – This is interesting, and possibly reflects a positive effect of human influence on annual plants. It might be interesting to investigate whether this exists in native floras, if it is a result of increased species introductions in human-modified landscapes, combined with a tendency for introduced species to be annuals. I also suspect there is some potential that the causality actually goes the other way – it may be that humans tend to prefer to live and work in landscapes that have many annual species.

Line 190 – This sounds a little alarmist in my opinion. It seems to imply that the very existence of perennial species is threatened. The results simply suggest that multiple factors will favor annuals in the future.

Line 322 – It’s not clear what “terms” refers to here.

Line 324 – This paragraph in general needs more elaboration. It’s not clear what constitutes an “entry”. I also don’t know how to interpret the idea of terms providing conflicting information. Does that mean conflicting with other terms, or that the description is somehow internally inconsistent? Again, a lack of clarity on what “term” means contributes to my confusion here.

Line 363 – I’m not sure what you mean “overcome rare species”. Does this just mean “exclude rare species”, and if so, why would that be a good goal?

Line 396 – It is not explicit, but I think the models used here are probably ordinary least squares regressions. This should be stated clearly. However, there are a few potential problems with standard linear regression here. First, the response variable is bounded between 0 and 1, so it might be prudent to use a logit link function in a glm (or some similar approach). Second, it is probable that there is residual spatial autocorrelation which could bias model parameters. The authors should explore this possibility with correlograms or a similar diagnostic and, if residual autocorrelation exists, implement a modeling approach that is not biased by it, such as a simultaneous autoregressive model.

Referee #3 (Remarks to the Author):

A. SUMMARY: The authors have pulled together a very significant dataset of life cycles (annual vs perennial), encompassing 235,000 of the extant plant species. They find that annual species are much less dominant across the planet than previously thought, but that they are found (as thought before) more frequently in hot and dry biomes.

B. ORIGINALITY: I found myself last month actually needing an estimate of how much of the Tree of Life is annual, and was shocked to see that the estimates are very old, and regional, rather than global. From that perspective alone, this paper is rather novel. However, I think that it has

scratched only the surface, and there's a lot more analytically that could be done to bring this to the expectation of a Nature publication. See below.

C & D. DATA AND STATS: The authors correlate percentage of annual species with various abiotic drivers such as temperature, precipitation, inter-annual variation in P and T, and inter-annual variation in P, as well as indexes from the human foot print database. While the paper is well written and the methods well explained, the authors have missed a unique opportunity in not having brought more explicit predictions of life history theory regarding the selection of annual vs perennial species. The methods do not take into account phylogenetic lack of independence, nor spatial autocorrelation. Instead, for the former, the authors repeated the analyses on the four better represented families. The results are likely to change once a more formal recognition of phylogenetic and spatial autocorrelations are taken into account.

E. With the methodological caveat I point out above regarding phylogenetic and spatially explicit analyses, the results seem appropriate. However, the authors' narrative assumes that plant species are not currently adapted to the types of environments in which they are found, and the expectation that annuals would have higher adult mortality in deserts than in wet ecosystems is not well substantiated in the intro or across the methods. The narrative about what these finds mean going forward is perhaps too far-stretched. To really talk with substance about what the future would bring, I would expect the authors to have carried out a macro-ecological population forecast worldwide, which is not what has been done here. Perhaps the authors are working on it as a follow up paper, but I'd argue that the simple report of where more annuals are found is not novel enough for Nature... where and when they will be found is more avant-guard.

G. REFERENCES: All seems correct.

H: CLARITY: It is unclear why the authors have decided to draw the artificial line of separating annual vs perennial where they have. We demographers and life history theoreticians know very well that that separation is artificial. There are many orders of magnitude in differences in longevity that are hidden in perennial species, and a couple of orders of magnitude in annuals. It seems like a really coarse approximation to collapse such diversity to short vs long-lived species, and this is not well explained. Likewise, the incorporation of the human foot print comes out of the blue, and it is not linked in any way to the hypotheses. On the topic of hypotheses, these assume that species are not locally adapted, and don't sufficiently draw from explanations from life history theory. likewise, in the exploration of inter-annual variation in P, I was not clear on what T was not also tested, nor why were moving windows not used, as we know that moving windows have recently shown that what happens on the same year or the previous year is not the best proxy of demographic rates - see work by Sanne Evers - <https://onlinelibrary.wiley.com/doi/10.1111/gcb.15519>

Minor comments:

L10. There are more than two kinds of life cycles in plants. This separation is too reductionist.

L13. Highlight why knowing these patterns is important

L16. Initially thought... by whom?

L17. Hot and dry is not new... what's the novelty?

L23. Change favourability here and elsewhere for selection

L25. Change are for may be

L43. Dramatically increased by how much?

L45. invasive plant species

L 50. Link of sentence starting "IN contrast..." is too abrupt

L47-54. Expectations are not clearly stated nor really solidly based on hypotheses from life history theory.

L 60. Benchmark 400 against the total number of plant species.

L 62. Perhaps talk here about the fact that there are two main kinds of deserts: warm and cold.

L 75. Do not say "enormous" - let the readership evaluate your efforts. Don't blow your own horn.

L 78. by life history theory

L80-81. This sentence implies that plants are not locally adapted.

L106-108. Comes out of the blue, and is not stand in the expectations in the intro.

L116. Phylogenetic comparative analyses are better approaches here

L149. And why not variation in T?

L154. HFP comes out of the blue, is not properly introduced, and not clearly articulated in the hypotheses. Also, no spatial autocorrelation tested?

L187-88. Unnecessarily poetic

L193-200. To really be able to finish on a paragraph like this, I would have expected this paper to carry out ecological forecasts, which it doesn't.

Other notes in the attached PDF.

Kind regards,
Rob Salguero-Gomez

**Author Rebuttals to Initial Comments:**

**Reviewer 1**

**Poppenwimer et al. compile and explore a comprehensive dataset on the distribution of**
**vascular plant species, combined with information on their life cycle strategies (annual vs**
**perennial). The dataset is then used to address fundamental questions on the global**
**biogeography of plant life cycle strategies, focusing on the importance of climate and**
**disturbance factors in shaping variation on the proportion of annual plant species. While I**
**find the manuscript interesting and with important results, there are major issues (listed**
**below) that need to be dealt with before the manuscript is suitable for consideration in**
**Nature.**

*We thank the reviewer for the overall positive feedback on our study and for the many*
*constructive suggestions.*

**Reasoning: The authors set out to test "key ecological hypotheses for plant life cycle**
**strategies that were never tested on a global scale". Which hypotheses? Biomes represent a**
**crucial component in the authors' analyses and results, but it is not clear how they fit into**
**the narrative. What are the key ecological (and evolutionary) hypotheses for plant life cycle**
**strategies across biomes? Should we expect similar patterns for similar biomes regardless**
**of geography? Higher proportion of annual plant species reflect the evolutionary rarity of**
**perenniality in biomes found under extreme climates?**

*We thank the reviewer for this constructive comment. We have restructured the *Introduction**
*section to better identify and portray our main hypotheses. Please see lines (51 – 60) & (83 – 90)*

*(lines 51-60): According to life-history theory, the optimal life cycle is determined by the*
*ratio of seedlings (or seeds) survival to adult survival^{25, 26}. The reproductive mode of perennials*
*requires multiple growing seasons⁵ compared to annuals which require only one growing*
*season. Therefore, any external condition that decreases the ability of plants to survive between*
*growing seasons necessarily reduces the reproductive fitness of perennial species^{25, 26}. However,*
*because annual species could survive such conditions as seeds rather than adults, their*

reproductive fitness may not be impacted⁵. Thus, any condition that skews the survivorship ratio
in favor of seeds should increase the favorability of annuals. Consequently, annuals should be
favored when adult mortality is high and seed persistence and seedling survival are relatively
high.

(lines 83 – 90): We tested three key hypotheses, predicting that annuals are favored
under: (1) increasing temperature and decreasing precipitation^{24, 33, 34, 35}, (2) high year-to-year
variability in climatic conditions^{35, 36, 37}, and (3) increasing human footprint (anthropogenic
disturbance^{36, 38, 39, 40}). All these hypotheses are based on the life-history theory that predicts
annual species to be favored with increasing adult mortality (relative to seedling mortality)^{25, 26}.
In other words, the relative abundance of annuals will be higher in regions with hot-dry
climates, high interannual variability, and disturbance because they decrease adult survival.

**Structure: Results and Discussion is almost entirely results, while the Conclusions section**
**reads as Discussion. I'd suggest bringing/adapting most of the text in the Conclusions to the**
**Results and Discussion section, and then have the Conclusions focused on the major**
**questions of the manuscript. Also, as I pointed out above, the authors need to make clear in**
**the Discussion which are the hypotheses being tested in the manuscript, and whether the**
**results support or reject each of them.**

We followed as suggested. We have incorporated much of the text from the original *Conclusion*
into the *Results and Discussion* section of the manuscript and further revised the *Results and*
*Discussion* section to better illustrate the results and indicate support for the presented
hypotheses. For example, in this section, we now directly reference our hypotheses:

(lines 105 – 106): *The variation in the annual-herb frequencies across biomes supports*
*the first hypothesis that annuals are favored with increasing temperature and lower precipitation*
*(Fig. 2).*

(lines 159 – 161): *We tested the second hypothesis that increased year-to-year climatic*
*variability favors annuals prevalence by focusing on interannual variability in total precipitation*
*(in terms of the coefficient of variation) and mean temperature (in terms of standard deviation).*

(lines 168 – 169): *We examined our third hypothesis that increased human footprint*
*(anthropogenic disturbance) should increase the proportion of annuals.*

Additionally, the *Conclusion* section now focuses on the overall broad impact of our study,
indicates the support and significance for our hypotheses, and emphasizes our key findings
without repeating the same material from the *Results and Discussion* section.

**Regressions: Have the authors checked for spatial autocorrelation (SAC) in model**
**residuals? These are not trivial results, and should be reported accordingly. If the authors**
**haven't checked for SAC, I'd suggest using (and presenting) correlograms, followed by (if**
**SAC is present) generalised least squares (GLS) analyses with different spatial structures**
**(e.g., spherical, exponential). Finally, the best model could be selected by comparing the**
**delta AIC of each model.**

Complete. We assessed the impact of spatial autocorrelation on our linear regression models as
suggested by the reviewer. The added analysis accounts for spatial autocorrelation (SAC) using
spatial eigenvectors (Dray *et al.* 2012). The results obtained using SAC are in full agreement
with the results obtained without SAC with very similar estimates of the model parameters.
These results are briefly summarized in the main text and are fully described in the supplement
(Supplement Note 2). However, since the SAC analysis artificially inflates the obtained R-
squared (the eigenvectors increased the proportion of explained variance) we focus on the
analysis without the spatial eigenvectors in the main text.

Dray, S., *et al.* Community ecology in the age of multivariate multiscale spatial analysis.
*Ecological Monographs*. **82**, 257-275 (2012).

**Phylogenetic context: The authors try to address a potential artifact of phylogenetic**
**dependence in the results by replicating some of the analyses using the four most annual-**
**rich families. Why not formally test for phylogenetic dependence in, for example, a pGLS**
**framework? The author could take advantage of published, comprehensive phylogenies**
**(e.g. Zanne *et al.* Nature, 2014; Smith and Brown AJB, 2018) and then add missing taxa**
**using imputation methods broadly used in the literature (e.g. SUNPLIN, randomized**
**within the most recent common ancestor).**

Complete. As suggested, we used the phylogeny of Smith and Brown, as the basis for these
analyses. The continuous response variable for each individual species was their median *mean*
*temperature of the warmest quarter* and *precipitation of the warmest quarter* across all their
observations. The explanatory variable was a numeric conversion of each species' life cycle; 1
for annual and 0 for perennial. We find that the results of pGLS further support our results and
refer to this analysis in the main text (lines 153 – 158). Full details of this analysis are presented
in the supplement (Supplement Note 4)

(lines 153 – 158): *Next, we tested the life cycle and climate relationship using*
*phylogenetic Generalized Least Squares (pGLS). We found that the median temperature of the*
*warmest quarter for annuals is 3°C higher, and the median precipitation of the warmest quarter*
*is 35% lower (Supplement Note 4). These results support the hypothesis that climate conditions*
*during the driest period play a significant role in driving the prevalence of annuals.*

**Biomes: Why use Whittaker's definition of biomes? Comparisons with previous studies is a**
**minor component of the manuscript, and the authors seem concerned with ambiguous**
**delimitations of terrestrial biomes (as expressed in the introduction). Despite being**
**climatically, compositionally and functionally distinct, many major biomes, such as**
**savannas (the second largest tropical biome), are lumped within one of Whittaker's biomes.**
**I'd suggest using Olson et al. (<https://academic.oup.com/bioscience/article/51/11/933/227116>)**
**or Higgins et al. for a functional perspective (<https://doi.org/10.1111/gcb.13367>).**

Following this comment, we further elaborated on the reasons why we defined biomes using
Whittaker's biomes classification (lines 451 – 453). First, Whittaker's biomes classification was
used in previous studies, which enables demonstrating the large differences in our estimation
compared with those found in textbooks.

(lines 451 – 453): *Whittaker's defined biomes were chosen to simplify comparisons to*
*previous estimates (the terminology used in textbooks best matched those of Whittaker's*
*definitions) and for its simplicity of using only temperature and precipitation.*

Second, Whittaker's biomes are only defined by *temperature* and *precipitation*. In contrast,
Olson's biomes have no clear definition, and Higgin's biomes are defined by vegetation height,
vegetation productivity index, and how vegetation activity is limited by temperature and soil
moisture. Consequently, the graphical representation of these biome definitions does not lend
itself to easy interpretation nor are such values readily available. Moreover, while it is reasonable
to assume that climate affects the vegetation, the directionality of the relationship between
vegetation attributes is unclear (does vegetation height affect life cycle or vice versa?)

**Spatial resolution: A key hypothesis in the manuscript is to test the importance of climatic**
**and landscape factors in shaping the biogeography of plant life cycle strategies globally.**
**Nonetheless, these factors are lumped into (very) coarse ecoregions, which encompass**
**heterogenous gradients of human disturbance and span multiple climatic domains. In the**
**Cerrado ecoregion (South American savannas), for instance, climate span from semi-arid**
**in its northeastern portion to sub-humid, frost affected regions in the south. Why not**
**assemble plant records into grid cells, which could then be classified into biomes according**
**to their overlap with Olson's (or Higgins) delimitatons. Issues with (i) grid cells being**
**assigned to wrong biomes (delimitation are no perfect in heterogenous landscapes,**
**especially in dry region of South America); (ii) low accuracy of geographic coordinates of**
**plant records, or (iii) species misidentifications could be dealt with by applying**
**randomizations and multiple iterations of the most relevant analyses.**

Complete. We reanalyzed the entire dataset using a gridded system (using square cells with
100km-by-100km). The results of the grid-based analysis are very similar to the results obtained
at the ecoregion level (Supplement Note 8 & Extended Data Fig 4) demonstrating the robustness
of our analyses and conclusions. Specifically, the linear regression model using annual climate
suggested annuals are more prevalent in hot and dry regions with the model using quarterly
climate once again producing a better fit. Similarly, climate unpredictability and human footprint
were both positively correlated with increased annual prevalence and their inclusion into the
quarterly climate model produced a better fitting model than without. After considering the pros
and cons of the ecoregion and grid-based, we chose to keep the ecoregion analysis in the main
text for two reasons:

(i) There is no logic dictating the layout of the gridded system. As such, there is no
attempt to reduce the overlap of ecosystems and therefore the results of the grid-based
approach were noisier (e.g., yearly precipitation and temperature explained 39%
rather than 48% of the variance). In contrast, by their definition (Olson *et al.* 2001),
ecoregions were explicitly designed to comprise “... *relatively large units of land or*
*water containing a distinct assemblage of natural communities sharing a large*
*majority of species, dynamics, and environmental conditions*”. Thus, when designing
ecoregions, an effort was made to produce ecoregions which limit the overlap of
various ecosystems where possible. Accordingly, many ecoregions are composed of
multiple smaller polygons and are not necessarily contiguous. Of the 825 ecoregions,
404 are composed of 5 or more distinct regions, and 158 are composed of 20 or more
distinct regions. For example, the Cerrado ecoregion in Brazil is composed of 29
distinct polygons with multiple ecoregions embedded within.

(ii) A large portion of the global map had insufficient data in the grid system (~ 60% of
the cells) as opposed to only 12% of ecoregions (Extended Data Fig 4A). A possible
method to overcome the insufficient data is to make the cells larger. However, this
introduces other biases. However, as the cells get larger, the likelihood of a single cell
covering multiple environmental conditions is greater. Similarly, because we limit our
cells to only those with 50% or more land coverage, bigger cells will more likely
exclude island regions, isthmuses, and peninsulas. For example, when increasing the
grid cell size to 150km-by-150km, the gridded system misses much of the Malay
Archipelago, Greek Islands, Sicily, and the Caribbean Islands (Extended Data Fig
4G).

Olson, D. M., *et al.* Terrestrial ecoregions of the world: a new map of life on Earth: a new global
map of terrestrial ecoregions provides an innovative tool for conserving biodiversity. *Bioscience*.
**51**, 933–938 (2001).

**Reviewer 2**

**Overall thoughts:**

**The fundamental quantification of the frequency of annual lifecycles in plants, and its**
**spatial variation is a useful product for a variety of purposes, ranging from evolutionary**
**studies to vegetation modeling. In fact, it is so fundamental that I assumed that this had**
**been done before. Some searching suggests that it has not, so this paper should fill an**
**important gap.**

**My main critique is that the paper is fairly uncritical of the data used. There are numerous**
**possible biases that could arise in the life cycle data and (especially) in the spatial data.**

**These could have large effects on the estimated frequencies. My guess is that the biases are**
**unlikely to overturn the major results here, but I think that needs some empirical support**
**in the manuscript (some suggestions are below). And I think it very plausible that such**
**biases could lead to errors in annual proportions of a few percentage points.**

**The statistical analysis of climate and annual proportion seemed mostly adequate, though I**
**did not find it massively illuminating. I think the main novelty in the paper rests firmly on**
**the map and biome-level patterns.**

*We thank the reviewer for the insightful comments and suggestions, which helped us to further*
*improve our study and the robustness of the conclusions. As detailed below, we have followed*
*the suggestions of the reviewer and conducted many of the analyses to verify the robustness of*
*our results to potential biases. We note that we have also conducted additional robustness*
*analyses following comments raised by the two other reviewers. Specifically, similar conclusions*
*were obtained when the analysis was conducted on a grid coordinate system instead of*
*ecoregions, and when the analysis accounted for the phylogenetic relationships among species*
*using pGLS. We believe that the entire set of analyses now provides strong support for the*
*validity of our conclusions.*

**Concerns about biases:**

**The central goal of the paper – quantification of annual proportions – depends on species**
**occurrences from GBIF. These data are not collected with the intent to obtain a**
**representative sample of the life cycles within a region, or in fact with any standardized**
**approach at all. Thus, there is a potential for important biases to arise, and these should be**
**investigated.**

**For example, individual plants may be sampled in GBIF approximately in proportion to**
**their abundance rather than in proportion to the richness of their life cycle category. This**
**will lead the more abundant group to be biased towards higher richness estimates. There**
**might also be important biases towards sampling more conspicuous species (likely favoring**
**perennials) and more consistently identifiable species (probably also favoring perennials).**
**Such potential biases should be investigated.**

**Some useful diagnostics would include species accumulation curves for annuals and**
**perennials within regions, and a plot of the relationship between number of GBIF records**
**and proportion annual (e.g. as a region becomes better sampled, does it tend to "drift"**
**towards higher or lower annual estimates)? Another approach that would help resolve**
**incomplete (and crucially, differently incomplete between annual and perennial) sampling**
**would be to apply a diversity estimator within each group within each region (such as the**
**Chao2 estimator). These estimated richnesses would not be free from all kinds of biases,**
**but at least some biases would be removed.**

**We completely agree that GBIF data may have inherent biases. Following this suggestion, we**
**assessed the relationships between the total number of GBIF observations in an ecoregion and**
**the proportion of annuals among all species and among herbaceous species and found no**
**correlation. In addition, we assessed the relationship between the total number of present (5+**
**observations) GBIF species in an ecoregion and the proportion of annuals in an ecoregion. This**
**analysis revealed no correlation when the analysis was conducted either among all species or**
**among herbaceous species. This analysis is detailed in the supplement (Supplement Note 9) and**
**extended data figure (Extended Data Fig 5).**

**The compilation of life cycle data is impressive, but it would be helpful to give some more**
**detail on its own potential biases. For example, what does the spatial pattern of missing**
**data look like (that is, where are there regions with many species with GBIF records but no**
**life cycle data?). How are the missing species distributed across the tree of life – are certain**
**clades over- or under-represented and do these clades tend to be annual- or perennial-**
**dominated?**

We explored the proportion of species missing from GBIF and the proportion missing from each
family. We found that our dataset is missing an average of approximately 17% of present (5+
observations) GBIF species per ecoregion (Supplement Note 9). Furthermore, we find no
identifiable hotspot areas of species lacking data. Additionally, we find no correlation between
the proportion of missing species from our dataset and the proportion of annuals. See supplement
for more details (Supplement Note 9) and extended data figure (Extended Data Fig 5).

Among the accepted species in the World Flora Online (excluding all families in Bryophyta), we
are missing, an average of approximately 22.2% of species per family (Extended Data Fig 5H).
For the great majority of families (393 out of 449), our dataset contains more than half of the
accepted species in the World Flora Online database. See supplement (Supplement Note 10).

**Line 27 – I think the phrase “broadest scale” is not quite right here. This implies something**
**about the scope or extent, whereas the main feature of this classification is its low**
**resolution, or coarseness.**

Completed. We have rephrased this on lines (31 – 33)

(lines 31 – 33): *Although crude, this categorization represents the most fundamental*
*characteristic of plant species and illustrates the inherent trade-offs between reproduction,*
*survival, and seedling success^{5, 7}.*

**Line 38-44 – I think this section is interesting in general, but did not really understand its**
**purpose here (especially when I later learned that crops were removed).**

Since *Nature* aims for a broad readership, we believe it is important to understand the relevance
of annuals and perennials in an agricultural context. Nonetheless, this part can be removed if the
editors argue it is unnecessary.

**Line 59 – I have not seen this estimate (13% annual) before. I agree that it is old and**
**seemingly based on too little data, but I’m not convinced that this is a widely accepted**
**number without better more recent alternatives. Has there really been no other estimate in**
**that time? For this section in general, I found the claims about the poor knowledge of**
**annual proportions to be shocking – thinking that there must better and more complete**
**evidence already published. I did try to find such numbers and my search turned up little,**
**so perhaps the situation is this bad, which certainly argues for the importance of this**
**manuscript!**

We thank the reviewer for highlighting the importance of our manuscript in filling this important
gap. We were also shocked at how poor the knowledge is. We conducted another comprehensive
literature review to find the numbers. Briefly, we did not find anything recent, but the number
13% is found many textbooks. After a lengthy investigation, we found that the original estimate
is about a century old from the classical work of Raunkiær (Extended Data Table Table 3 and
Extended Data Table 4).

**Line 187-189 – This is interesting, and possibly reflects a positive effect of human influence**
**on annual plants. It might be interesting to investigate whether this exists in native floras, if**
**it is a result of increased species introductions in human-modified landscapes, combined**
**with a tendency for introduced species to be annuals. I also suspect there is some potential**
**that the causality actually goes the other way – it may be that humans tend to prefer to live**
**and work in landscapes that have many annual species.**

Following this comment, we investigated whether the climatic conditions that favor annuals are
correlated with a human footprint. By applying linear regression, we found that both yearly
temperature and precipitation model and quarterly temperature and precipitation were weakly
correlated with human footprint (Supplement Note 11 and Extended Data Fig 6). However, we

should note that while we cannot completely rule out this option within the context of
observational study, it seems unlikely.

**Line 190 – This sounds a little alarmist in my opinion. It seems to imply that the very**
**existence of perennial species is threatened. The results simply suggest that multiple factors**
**will favor annuals in the future.**

First, we now include an additional analysis that better illustrates the predicted change in the
worldwide life-from distribution. This analysis is based on a linear regression between the two-
most influential climatic parameters found in our study (mean temperature and precipitation
during the warmer quarter) and predicted climates in 2100. This back-of-the-envelope projection
illustrates that, all else being equal, the mean annual proportion per ecoregion could be as high as
32%. This analysis is now presented in lines (174 – 178) and in Extended Data Fig 3.

(lines 174 – 178): *Finally, we built a back-of-the-envelope projection of the expected*
*prevalence of annuals in 2100 based on predicted changes in mean temperature and*
*precipitation at the warmest quarter⁴² (Extended Data Fig 3). Under the simplifying assumptions*
*that the prevalence of annuals in the future will follow the same climatic patterns without*
*adaptation or time-lag, our model suggests that ~81% of ecoregions will experience an increase*
*in the proportion of annuals.*

Second, we have rephrased the mentioned sentence on lines (212 – 215)

(lines 212 – 215): *With the human population predicted to reach 11 billion by 2100,*
*anthropogenic activities are expected to play an increasing role in shaping patterns of plant*
*biogeography. Consequently, we expect a world with more annual-dominated ecoregions.*

**Line 322 – It’s not clear what “terms” refers to here.**

**Line 324 – This paragraph in general needs more elaboration. It’s not clear what**
**constitutes an “entry”. I also don’t know how to interpret the idea of terms providing**
**conflicting information. Does that mean conflicting with other terms, or that the**

**description is somehow internally inconsistent? Again, a lack of clarity on what “term”**
**means contributes to my confusion here.**

We have included a definition of an *entry* and added a few examples of what we now call *trait*
*terms* to better illustrate our methods. In lines (352 – 359) we write:

(lines 352 – 359): *Following name resolution, each entry consisted of a single species*
*name and its associated trait term (e.g., annual, forb/herb, tree, 10 years, Shrub/Herb, aquatic,*
*TREE, epiphyte, etc.). All unique trait terms were manually assessed to extract data relevant to a*
*plant’s life cycle (annual/perennial) and growth form (woody/herbaceous) when available.*
*Those that did not provide relevant information (e.g., Terrestrial_Trailing_Plant, 2.4, NO, b H,*
*etc.) or provided conflicting information for the same entry were excluded (e.g., Shrub/Herb,*
*Tree/Terrestrial Herb, etc.). After term interpretation, there were 5.6 million entries and 262,000*
*unique species remaining.*

**Line 363 – I’m not sure what you mean “overcome rare species”. Does this just mean**
**“exclude rare species”, and if so, why would that be a good goal?**

We have better explained our reasoning for the 5-observation threshold for species presence. See
lines (395 – 397).

(lines 395 – 397): *Following the procedures used by¹², species were only considered*
*“present” in a geographic region if there were five or more observations to ensure the species*
*had a sufficient established population.*

**Line 396 – It is not explicit, but I think the models used here are probably ordinary least**
**squares regressions. This should be stated clearly. However, there are a few potential**
**problems with standard linear regression here. First, the response variable is bounded**
**between 0 and 1, so it might be prudent to use a logit link function in a glm (or some**
**similar approach).**

We thank the reviewer for this suggestion. Indeed, there are many approaches for analyzing
proportions, including a standard linear regression, a standard regression using transformations

(logit), and several types of GLM. The main problems with the standard regression are found
only in small datasets and are irrelevant for our large sample size. Moreover, a recent
comparison (Ferrari *et. al* 2016) has concluded that “Overall, linear regression can be an
acceptable method to perform hypothesis testing with proportion data, ... being quite robust to
model misspecification. It is in general more conservative than mixed binomial or beta-binomial
models”. Lastly, unlike GLM, the standard regression provides an R-squared, which we believe
is an important metric to demonstrate the explanatory power of the models.

Nonetheless, to further test the robustness of our conclusion, we also investigated the data after a
logit transformation and a GLM with a Poisson link function. For both found very similar results
(Supplement Note 3 and Extended Data Fig 2).

Ferrari, A., Comelli., M. A comparison of methods for the analysis of binomial clustered
outcomes in behavioral research. *Journal of Neuroscience Methods*. **274**, 131-140 (2016).

**Second, it is probable that there is residual spatial autocorrelation which could bias model**
**parameters. The authors should explore this possibility with correlograms or a similar**
**diagnostic and, if residual autocorrelation exists, implement a modeling approach that is**
**not biased by it, such as a simultaneous autoregressive model.**

We have added an analysis that accounts for spatial autocorrelation using spatial eigenvectors¹
and found no difference in the parameter estimation. However, since this analysis artificially
inflates the R-squared (incorporating multiple eigenvectors artificially increases the proportion of
variance explained), we present the analysis without the spatial eigenvectors in the main text and
refer the readers to the supplement (Supplement Note 2) for the results that incorporate the
spatial autocorrelation.

**Reviewer 3**

**A. SUMMARY: The authors have pulled together a very significant dataset of life cycles**
**(annual vs perennial), encompassing 235,000 of the extant plant species. The find that**
**annual species are much less dominant across the planet than previously thought, but that**
**they are found (as thought before) more frequently in hot and dry biomes.**

**B. ORIGINALITY: I found myself last month actually needing an estimate of how much of**
**the Tree of Life is annual, and was shocked to see that the estimates are very old, and**
**regional, rather than global. From that perspective alone, this paper is rather novel.**
**However, I think that it has scratched only the surface, and there's a lot more analytically**
**that could be done to bring this to the expectation of a Nature publication. See below.**

**C & D. DATA AND STATS: The authors correlate percentage of annual species with**
**various abiotic drivers such as temperature, precipitation, inter-annual variation in P and**
**T, and inter-annual variation in P, as well as indexes from the human foot print database.**
**While the paper is well written and the methods well explained, the authors have missed a**
**unique opportunity in not having brought more explicit predictions of life history theory**
**regarding the selection of annual vs perennial species.**

We thank the reviewer for highlighting the importance of our manuscript and for the many
constructive suggestions. We have rephrased the *Introduction* section to highlight the key
hypotheses of life history theory that are being examined in lines (51 – 60) and (83 – 90).

(lines 51 – 60): *According to life-history theory, the optimal life cycle is determined by*
*the ratio of seedlings (or seeds) survival to adult survival^{25, 26}. The reproductive mode of*
*perennials requires multiple growing seasons⁵ compared to annuals which require only one*
*growing season. Therefore, any external condition that decreases the ability of plants to survive*
*between growing seasons necessarily reduces the reproductive fitness of perennial species^{25, 26}.*
*However, because annual species could survive such conditions as seeds rather than adults,*
*their reproductive fitness may not be impacted⁵. Thus, any condition that skews the survivorship*
*ratio in favor of seeds should increase the favorability of annuals. Consequently, annuals should*

*be favored when adult mortality is high and seed persistence and seedling survival are relatively*
*high.*

(lines 83 – 90): *We tested three key hypotheses, predicting that annuals are favored*
*under: (1) increasing temperature and decreasing precipitation^{24, 33, 34, 35}, (2) high year-to-year*
*variability in climatic conditions^{35, 36, 37}, and (3) increasing human footprint (anthropogenic*
*disturbance^{36, 38, 39, 40}). All these hypotheses are based on the life-history theory that predicts*
*annual species to be favored with increasing adult mortality (relative to seedling mortality)^{25, 26}.*
*In other words, the relative abundance of annuals will be higher in regions with hot-dry*
*climates, high interannual variability, and disturbance because they decrease adult survival.*

**The methods do not take into account phylogenetic lack of independence, nor spatial**
**autocorrelation. Instead, for the former, the authors repeated the analyses on the four**
**better represented families. The results are likely to change once a more formal recognition**
**of phylogenetic and spatial autocorrelations are taken into account.**

Following this comment, we have incorporated a pGLS analysis to assess the impact of
phylogenetic relatedness. The continuous response variable for each individual species was their
median *temperature of the warmest quarter* and *precipitation of the warmest quarter*. The
explanatory variable was the species' life cycle categorization (1 for annuals and 0 for
perennials). The results of this analysis indicated that annuals are found in regions with warmer
temperatures and lower precipitation during the summer and further supported our conclusions.
This analysis is now in the main text (lines 153 – 158) and is fully described in the supplement
(Supplement Note 4).

(lines 153 – 158): *Next, we tested the life cycle and climate relationship using*
*phylogenetic Generalized Least Squares (pGLS). We found that the median temperature of the*
*warmest quarter for annuals is 3°C higher, and the median precipitation of the warmest quarter*
*is 35% lower (Supplement Note 4). These results support the hypothesis that climate conditions*
*during the driest period play a significant role in driving the prevalence of annuals.*

Second, we assessed the impact of spatial autocorrelation on our linear regression models as
suggested by the reviewer. The added analysis accounts for spatial autocorrelation using spatial

eigenvectors (Dray *et al.* 2012). Using this correction, there was virtually no difference in the
parameter estimation compared to the analysis without it. However, since this analysis artificially
inflates the R-squared (the eigenvectors increased the proportion of explained variance) we focus
on the analysis without the spatial eigenvectors in the main text, while providing the full details
of the analysis with the spatial autocorrelation in the appendix (Supplement Note 2).

Dray, S., *et al.* Community ecology in the age of multivariate multiscale spatial analysis.
*Ecological Monographs*. **82**, 257-275 (2012).

**E. With the methodological caveat I point out above regarding phylogenetic and spatially**
**explicit analyses, the results seem appropriate. However, the authors' narrative assumes**
**that plant species are not currently adapted to the types of environments in which they are**
**found, and the expectation that annuals would have higher adult mortality in deserts than**
**in wet ecosystems is not well substantiated in the intro or across the methods.**

**The narrative about what these finds mean going forward is perhaps too far-stretched. To**
**really talk with substance about what the future would bring, I would expect the authors to**
**have carried out a macro-ecological population forecast worldwide, which is not what has**
**been done here. Perhaps the authors are working on it as a follow up paper, but I'd argue**
**that the simple report of where more annuals are found is not novel enough for Nature...**
**where and when they will be found is more avant-guard.**

We thank the reviewer for this suggestion. Making a robust projection regarding the future
distribution of plant life cycles requires comprehensive data that are currently not available on a
global scale. Specifically, such models necessitate population-level data and a species interaction
matrix that includes most plant species on earth. However, following this comment, we have
built a back-of-the-envelope model with predictions of the future prevalence of annuals based on
the projected climate in the year 2100 and the current climatic niches of annuals and perennials,
as found in our study. While we acknowledge that, as with all forecasts, this model is based on
very simplifying assumptions, it predicts that ~81% of ecoregions will experience an increase in
the proportion of annual herbs. See lines (174 – 178) and figure (Extend Data Fig 1).

(lines 174 – 178): *Finally, we built a back-of-the-envelope projection of the expected*
*prevalence of annuals in 2100 based on predicted changes in mean temperature and*
*precipitation at the warmest quarter^{A2} (Extended Data Fig 3). Under the simplifying assumptions*
*that the prevalence of annuals in the future will follow the same climatic patterns without*
*adaptation or time-lag, our model suggests that ~81% of ecoregions will experience an increase*
*in the proportion of annuals.*

**G. REFERENCES: All seems correct.**

**H. CLARITY: It is unclear why the authors have decided to draw the artificial line of**
**separating annual vs perennial where they have. We demographers and life history**
**theoreticians know very well that that separation is artificial. There are many orders of**
**magnitude in differences in longevity that are hidden in perennial species, and a couple of**
**orders of magnitude in annuals. It seems like a really coarse approximation to collapse**
**such diversity to short vs long-lived species, and this is not well explained.**

We agree that the classification of longevity into annual and perennials inevitably involves loss
of information. Nonetheless, despite being crude, this dichotomous classification is widely
acknowledged by botanists and the wider scientific community and was helpful in various
disciplines, including ecology, evolution, biogeography, and agriculture (e.g., the reviewer
mentioned above that they needed to know how much the tree of life is annual). Importantly, the
life cycle attribute of annual/perennial is available for the great majority of species on earth,
while data on longevity is scarce; therefore, this classification is the best we can presently obtain.
Still, we agree with the reviewer that differentiating between the plant lifespan using a
continuous scale holds great promise for future work, but it will necessitate a collaborative
community effort to collect and assemble. We, therefore, added a sentence about the limitation
of this common classification (lines 31 – 33).

(lines 31 – 33): *Although crude, this categorization represents the most fundamental*
*characteristic of plant species and illustrates the inherent trade-offs between reproduction,*
*survival, and seedling success^{5, 7}.*

**Likewise, the incorporation of the human foot print comes out of the blue, and it is not**
**linked in any way to the hypotheses.**

Complete. We rephrased the introduction section to better present the connection between
anthropogenic disturbance and the relative frequency of annuals. In lines (57 – 58) and (88 – 90)
we write:

*(lines 57 – 58): Thus, any condition that skews the survivorship ratio in favor of seeds*
*should increase the favorability of annuals. Consequently, annuals should be favored when adult*
*mortality is high and seed persistence and seedling survival are relatively high.*

*(lines 88 – 90): In other words, the relative abundance of annuals will be higher in*
*regions with hot-dry climates, high interannual variability, and disturbance because they*
*decrease adult survival.*

**On the topic of hypotheses, these assume that species are not locally adapted, and don't**
**sufficiently draw from explanations from life history theory.**

Completed. In the revised manuscript we elaborate about the hypotheses. Indeed, life history
theory does not explicitly relate to environmental conditions, which necessitates incorporating
additional assumptions about mortality under different conditions. However, we did not claim
that “species are not locally adapted”. Instead, our logic is based on a more conservative
assumption that local adaptations are not absolute (while a desert plant can be more adapted to
drought than a plant that never experiences water stress, it is still more likely to experience
drought-related mortality).

**likewise, in the exploration of inter-annual variation in P, I was not clear on what T was**
**not also tested,**

Completed. Following this comment, we have incorporated an interannual temperature
variability metric (like the precipitation metric already included) into the analyses. We found that
increasing temperature variability also increases the favorability of annuals, though its effect is
very weak. This result is now detailed in lines (163 – 165).

(lines 163 – 165): *Likewise, we found that increasing temperature variability also*
*increases the favorability of annuals, though its effect is much weaker ($P = 0.0003$, $D.F. = 679$,*
*$R^2 = 0.02$).*

**nor why were moving windows not used, as we know that moving windows have recently**
**shown that what happens on the same year or the previous year is not the best proxy of**
**demographic rates - see work by Sanne Evers -**

**<https://onlinelibrary.wiley.com/doi/10.1111/gcb.15519>**

We agree that moving windows are a better proxy than just using data from a single year.
However, none of our climatic features were produced using a single year's worth of data. The
historical climate data were developed from climate data from 1970 – 2000. Similarly, our
analyses also include inter-annual variability for both precipitation and temperature measures as
recorded in the years 1961-2018. We have revised our description of their construction to make it
clear that these metrics use 58 years of climate measures. In lines (413 – 414) and lines (421 –
425) we write:

(lines 413 – 414): *We downloaded bioclimate features from the WorldClim Global*
*Climate Data⁶⁰, which were developed from climate data during 1970 – 2000, at ten arc-minutes*
*resolution.*

(lines 421 – 425): *We aggregated all available monthly precipitation/temperature data*
*layers from the WorldClim Global Climate Data⁶⁰ at ten arc-minutes resolution (1961 – 2018) to*
*determine the total yearly precipitation and mean temperature for each pixel in each year. The*
*mean and coefficient of variation of the total yearly precipitation across all 58 years for each*
*pixel were then used to determine IPV values.*

**Minor comments:**

**L10. There are more than two kinds of life cycles in plants. This separation is too**
**reductionist.**

We agree. Please see our reply above.

**L13. Highlight why knowing these patterns is important**

In the revised manuscript we expand on the various hypotheses concerning the relative
distribution of annuals and perennials, their evolutionary trade-offs, on the association between
life-cycle strategies on central physiological traits, and their relevance to agriculture.

Unfortunately, however, we cannot expand on these topics in the Abstract due to the strict 200
words limit. We believe the Abstract already showcases the main findings of our study and that
readers will appreciate the importance of our results by reading the revised Introduction.

**L16. Initially thought... by whom?**

Fixed. references were added.

**L17. Hot and dry is not new... what's the novelty?**

We have rephrased this part to make it clearer that our results reveal a more delicate pattern,
whereby the prevalence of annuals is driven by temperature and precipitation in the driest
quarter, rather than yearly means. Still, within the 200-word limit of the abstract, we cannot
expand on this and compare the findings to previous knowledge. However, within the manuscript
itself we explain that previous knowledge is only about desert biomes having more annuals,
while our analysis highlights the role of hot temperatures as well as the timing of the
precipitation (e.g., why some Mediterranean systems have more annuals than deserts).

**L23. Change favourability here and elsewhere for selection**

We thank the reviewer for this suggestion. However, we prefer the term “favored” in lieu of
selection because, for example, in hot and dry regions annuals are not selected for but are instead
more favored than in other regions.

**L25. Change are for may be**

This sentence has been removed from the revised manuscript.

**L43. Dramatically increased by how much?**

Fixed. We clarified that ~70% of global cropland are annual species. See lines (46 – 47)

(lines 46 – 47): *Annual plants cover ~70% of the croplands and provide ~80% of*
*worldwide food consumption*²².

**L45. invasive plant species**

Fixed as suggested.

**L 50. Link of sentence starting “IN contrast...” is too abrupt**

This paragraph has been entirely changed in the revised manuscript.

**L47-54. Expectations are not clearly stated nor really solidly based on hypotheses from life**
**history theory.**

Done. We restructured the entire Introduction to better present the hypotheses and expectations.

Additionally, these are now summarized in the final paragraph of the Introduction (lines 86 –
90).

(lines 86 – 90): *All these hypotheses are based on the life-history theory that predicts*
*annual species to be favored with increasing adult mortality (relative to seedling mortality)*^{25, 26}.

*In other words, the relative abundance of annuals will be higher in regions with hot-dry*
*climates, high interannual variability, and disturbance because they decrease adult survival.*

**L 60. Benchmark 400 against the total number of plant species.**

Done. We now make it clear that 400 species represents approximately 0.1% of accepted
vascular plants. See lines (64 – 66)

(lines 64 – 66): *First, the current estimate for the global proportion of annual species*
*(13%) is based on a century-old sample of merely 400 species⁶, representing 0.1% of accepted*
*plant species²⁷.*

**L 62. Perhaps talk here about the fact that there are two main kinds of deserts: warm and**
**cold.**

We have mentioned that there are two main kinds of deserts in lines (71 – 73)

(lines 71 – 73): *Lastly, each biome incorporates a wide range of conditions, e.g., the*
*mean temperature in the desert biome ranges from 30° to -10°C, corresponding to hot and cold*
*deserts.*

**L 75. Do not say "enormous" - let the readership evaluate your efforts. Don't blow your**
**own horn.**

Fixed as suggested.

**L 78. by life history theory**

Fixed as suggested in this sentence and in few others (lines 86 – 90).

(lines 86 – 90): *We tested three key hypotheses, predicting that annuals are favored*
*under: (1) increasing temperature and decreasing precipitation^{24, 33, 34, 35}, (2) high year-to-year*
*variability in climatic conditions^{35, 36, 37}, and (3) increasing human footprint (anthropogenic*
*disturbance^{36, 38, 39, 40}). All these hypotheses are based on the life-history theory that predicts*
*annual species to be favored with increasing adult mortality (relative to seedling mortality)^{25, 26}.*
*In other words, the relative abundance of annuals will be higher in regions with hot-dry*
*climates, high interannual variability, and disturbance because they decrease adult survival.*

**L80-81. This sentence implies that plants are not locally adapted.**

As replied above, we did not claim that “species are not locally adapted”, but on a more
conservative assumption that local adaptations are not absolute.

**L106-108. Comes out of the blue and is not stand in the expectations in the intro.**

Fixed. We rephrased the introduction to make the hypotheses clearer. Lines (75 – 92).

**L116. Phylogenetic comparative analyses are better approaches here**

As detailed in our response above, we have conducted complementary pGLS analyses to account
for the phylogenetic relationships among species (lines 153 – 156). The results obtained using
these analyses confirmed all our conclusions.

(lines 153 – 156): *Next, we tested the life cycle and climate relationship using*
*phylogenetic Generalized Least Squares (pGLS). We found that the median temperature of the*
*warmest quarter for annuals is 3°C higher, and the median precipitation of the warmest quarter*
*is 35% lower (Supplement Note 4).*

**L149. And why not variation in T?**

Done. As detailed in our response above, we have incorporated interannual temperature
variability, see lines (163 – 165).

(lines 163 – 165): *Likewise, we found that increasing temperature variability also*
*increases the favorability of annuals, though its effect is much weaker ($P = 0.0003$, $D.F. = 679$,*
*$R^2 = 0.02$).*

**L154. HFP comes out of the blue, is not properly introduced, and not clearly articulated in**
**the hypotheses. Also, no spatial autocorrelation tested?**

We have altered the Introduction to better introduce human footprint as a feature being assessed,
see lines (88 – 90). Similarly, we have incorporated spatial autocorrelation analyses in the
supplement (Supplement Note 2).

(lines 88 – 90): *In other words, the relative abundance of annuals will be higher in*
*regions with hot-dry climates, high interannual variability, and disturbance because they*
*decrease adult survival.*

**L187-88. Unnecessarily poetic**

This sentence has been removed considering the included future projection analysis.

**L193-200. To really be able to finish on a paragraph like this, I would have expected this**
**paper to carry out ecological forecasts, which it doesn't.**

We have conducted a future projection analysis and have changed this paragraph as a result. This
analysis is presented and discussed in lines (174 – 178).

(lines 174 – 178): *Finally, we built a back-of-the-envelope projection of the expected*
*prevalence of annuals in 2100 based on predicted changes in mean temperature and*
*precipitation at the warmest quarter⁴² (Extended Data Fig 3). Under the simplifying assumptions*
*that the prevalence of annuals in the future will follow the same climatic patterns without*
*adaptation or time-lag, our model suggests that ~81% of ecoregions will experience an increase*
*in the proportion of annuals.*

**Figure 1. Font size too small [referencing histogram]**

Fixed. The font size of the embedded histogram has been increased.

**Figure 1. Chose another color – does not pop [referencing the grey color in map]**

Fixed. The grey color been changed to a darker grey.

Reviewer Reports on the First Revision:

Referees' comments:

Referee #1 (Remarks to the Author):

Dear Dr. Poppenwimer et al.:

I have now read your revised MS, and checked the revisions against the comments from the other two referees and myself. You have handled the comments in a sound manner, which improved clarity in the new version. The MS is now scientifically more robust, and I will therefore recommend it for publication.

A final, minor comment: The authors account for spatial autocorrelation in the new version of the MS, which is appropriate. But despite the presence of SAC, the authors decided not to include spatial vectors in the final models because "SAC analysis artificially inflates the obtained R-squared". Yet, the presence of SAC will affect estimates of F and, consequently, P values (the latter are presented as results in the MS). To avoid this, the authors could perform variation partitioning analyses and then report adj. r-squared values that are independent of SAC, as well as r-squared values attributed to the spatially-structured component of the explanatory variables.

Kind regards,
Danilo Neves

Referee #2 (Remarks to the Author):

This is my second review of this paper. I remain convinced of its importance and novelty. In this revision, the authors have provided the additional analyses required to convince me of the robustness of data and statistical approaches. I have no remaining major concerns.

One small comment:

Extended Data Figure 6: I think the caption description doesn't match the panel labeling. Panels A,C,E seem to show what is described as panels A-C.

Referee #3 (Remarks to the Author):

This is the second time that I'm reviewing this work. The topic remains as important and timely as in the previous version, and the quality of the writing, and focus around testing specific life history theory predictions has improved drastically. The quality of the writing is very high, and all of the analyses have been performed to a very high standard. I appreciate, however, just how difficult it is to project the percentage of annual life histories into the future. I remain convinced that this part of the exercise is critical to better contextualise and time-prove this paper. However, the details about how this has been done are too light for me to evaluate the soundness of the approach. The authors themselves acknowledge a "back-of-the-envelope" approach, but I need to know exactly what was done (the details in at the end of the methods are not sufficient). As the projection makes a number of assumption (not all of which are detailed in the paper, but are in the responses from the authors to the last round of reviews), I'd suggest shortening the projection interval from 2100 to perhaps 20-30 years (e.g. 2040-50) so the uncertainty is less propagated/smaller.

Pending on this important aspect (the projection, and some minor aspects annotated in the

attached), this is an outstanding paper.

Importantly, given the clarity and importance of the results, the title is too vague. Why not changing it to something like "There are fewer annual plant species worldwide than previously thought". Plant life cycles implies many other life history traits that the authors do not talk about in this piece of research.

I hereby give up my right of anonymity. I hope that these comments are useful to the authors.
Rob Salguero-Gomez

**Author Rebuttals to First Revision:**

**Reviewer 1**

**I have now read your revised MS, and checked the revisions against the comments from the**
**other two referees and myself. You have handled the comments in a sound manner, which**
**improved clarity in the new version. The MS is now scientifically more robust, and I will**
**therefore recommend it for publication.**

We thank the reviewer for their helpful suggestions and appreciate their overall positive
recognition of the revisions.

**A final, minor comment: The authors account for spatial autocorrelation in the new**
**version of the MS, which is appropriate. But despite the presence of SAC, the authors**
**decided not to include spatial vectors in the final models because "SAC analysis artificially**
**inflates the obtained R-squared". Yet, the presence of SAC will affect estimates of F and,**
**consequently, P values (the latter are presented as results in the MS). To avoid this, the**
**authors could perform variation partitioning analyses and then report adj. r-squared**
**values that are independent of SAC, as well as r-squared values attributed to the spatially-**
**structured component of the explanatory variables.**

We thank the reviewer for this suggestion. Following the comment of the reviewer on the
previously submitted version, we have performed an analysis that accounts for spatial
autocorrelation. We report the results in the main text of the manuscript (lines 119 – 122) and
provide a detailed description of this analysis in the main text of the manuscript (lines 519 – 524)
and in Supplement Note 3. Given the large sample size ($n = 723$), the adjusted r-squared value
obtained in this analysis was only marginally smaller than that obtained in the original analysis.
Importantly, the main problem with reporting the results with spatial vectors in the main text is
that including them artificially inflates the R-squared, so instead of describing the proportion of
variance explained by our model (e.g., rainfall, temperature), it also reflects the effect of the
spatial vectors. Such a problem cannot be overcome by using adjusted R-squared, which adjusts
for the number of variables, not for parts of the model that is only used to account for
independence among observations. Taken together, we prefer to keep the analysis without the

- spatial vectors in the main text and provide the detailed description of the analysis with the
- spatial vectors in the appendix.

**Reviewer 2**

**This is my second review of this paper. I remain convinced of its importance and novelty.**
**In this revision, the authors have provided the additional analyses required to convince me**
**of the robustness of data and statistical approaches. I have no remaining major concerns.**

*We thank the reviewer for their supportive feedback on our study.*

**One small comment: Extended Data Figure 6: I think the caption description doesn't**
**match the panel labeling. Panels A,C,E seem to show what is described as panels A-C.**

*We have updated the labels of the panels in Extended Data Figure 6 and the corresponding*
*caption to ensure that all descriptions accurately align.*

**Kind regards,**

**Danilo Neves**

**Reviewer 3**

**This is the second time that I'm reviewing this work. The topic remains as important and**
**timely as in the previous version, and the quality of the writing, and focus around testing**
**specific life history theory predictions has improved drastically. The quality of the writing**
**is very high, and all of the analyses have been performed to a very high standard. I**
**appreciate, however, just how difficult it is to project the percentage of annual life histories**
**into the future. I remain convinced that this part of the exercise is critical to better**
**contextualise and time-prove this paper.**

We greatly appreciate the support for the validity and importance of this manuscript.

**However, the details about how this has been done are too light for me to evaluate the**
**soundness of the approach. The authors themselves acknowledge a "back-of-the-envelope"**
**approach, but I need to know exactly what was done (the details in at the end of the**
**methods are not sufficient). As the projection makes a number of assumption (not all of**
**which are detailed in the paper, but are in the responses from the authors to the last round**
**of reviews), I'd suggest shortening the projection interval from 2100 to perhaps 20-30 years**
**(e.g. 2040-50) so the uncertainty is less propagated/smaller.**

We thank the reviewer for this suggestion. First, we have updated our projections by
incorporating anticipated climate estimates for the period from 2041 to 2060. Due to the inherent
uncertainty of climate predictions, we present our projection conservatively by focusing on the
latter part of this range (i.e., 2060).

Second, we have expanded the description of our model in the Results and Discussion section
lines (176 – 178) and included more details in the Methods section (lines 533– 543). In the
revised version we now write:

*(lines 176 – 178): Under the simplifying assumptions that the prevalence of annuals in*
*the future will follow the same climatic patterns without adaptation or time-lag, our model*
*suggests that ~69% of ecoregions will experience an increase in the proportion of annuals.*

(lines 533 – 543) *To obtain future projections of the proportion of annual herbs in each*
*ecoregion, future climate estimates in the year 2060 were downloaded from the WorldClim*
*Global Climate Data⁵⁹ using the 2041-2060, UKESM1-0-LL⁶⁵, ssp585, at ten arc-minutes*
*resolution. The median values for each bioclimatic variable were extracted for each ecoregion*
*using the R package raster v3.4-13⁵⁷. Using the coefficients of a linear regression between the*
*two-most influential climatic parameters found in our study (mean temperature and precipitation*
*during the warmer quarter, i.e., the Quarterly-Model) and their predicted median value in each*
*ecoregion in 2060, we produced estimates for the proportion of annual herbs in each ecoregion*
*with sufficient data. Year-to-year climate variability and human footprint were not incorporated*
*due to data unavailability at the required resolution and scale. The projected annual herbs*
*proportion in each ecoregion was compared to its current estimate to determine the predicted*
*change in proportion.*

**Pending on this important aspect (the projection, and some minor aspects annotated in the**
**attached), this is an outstanding paper.**

We thank the reviewer for the kind words.

**Importantly, given the clarity and importance of the results, the title is too vague. Why not**
**changing it to something like "There are fewer annual plant species worldwide than**
**previously thought". Plant life cycles implies many other life history traits that the authors**
**do not talk about in this piece of research.**

We agree that, in many cases, the preferable title summarizes the main take-home message of the
paper. However, in our particular case, a main strength is being comprehensive with several
main findings. We have thought about titles such as “There are fewer annuals species than
previously thoughts,” or “Annuals are favored by hot-dry summer”, and “Supporting the life-
history theory”, but we feel that they will undermine the significance of our findings which
cannot be summarized in one sentence.

Nonetheless, following Dr. Salguero-Gomez's comment that "plant life cycles" is unclear, we
have updated the title to *Revising the global biogeography of annual and perennial plants*.

**I hereby give up my right of anonymity. I hope that these comments are useful to the**
**authors.**

**Rob Salguero-Gomez**

**Minor comments:**

**L1. Title is not very telling of the research finding.**

We have revised the title, see above for full description.

**L13. Replace "which" with "that"**

Replaced

**L15. Add "their"**

Added

**L18: add "plant"**

Added

**L27. Add comma**

Added

**L50. Could you state how many approximately?**

We added an estimate for the number of families that have at least one annual plant in lines (50 –
51).

(lines 50 – 51): *The annual life cycle has repeatedly evolved in at least 120 different*
*families, suggesting that it provides a fitness advantage under certain conditions*²⁴.

**L75. Remove “plant”**

Removed

**L79. Replace “the largest” with “a comprehensive”**

Replaced

**L82. Remove “further” and allow”ed”**

Removed

**L95-96. Not a result! Move up. (referencing text: *We have assembled an extensive plant***
***growth-form database containing life cycle data for 67% of all vascular plant species and***
***georeferenced data for 51%. ”?***

This text has been moved into the final paragraph of the introduction.

(lines 82 – 83): *This extensive plant growth-form database contains life cycle data for*
*67% of all vascular plant species and georeferenced data for 51%.*

**L103. Remove “very”**

Removed

**L103. Noun missing**

Added missing noun “results”.

**L153. Of...**

Continued the sentence.

(lines 152 – 154): *Qualitatively similar relationships between climate and annual*
*proportion were found in all families (Fig 4), providing evidence for convergent evolution of*
*annual life cycles in hot and dry conditions.*

**L178. Add + details of simplifying assumptions**

Added more text better detailing the assumptions of the model in lines (176 – 178).

**L213. Perhaps project only by 30 years**

The projected annual herb estimates now uses the estimated bioclimatic measures for years 2041
186 – 2060 instead of 2081 – 2100. As such, we have provided an estimate for human population in
2060 rather than 2100.

(line 212 – 214): *With the human population predicted to reach 10 billion by 2060,*
*anthropogenic activities are expected to play an increasing role in shaping patterns of plant*
*biogeography.*

**L538-544. Not enough details**

Done. More details are now provided on this analysis. These are now provided in lines (533 –
543)